# Systematic discovery of protein interaction interfaces using AlphaFold and experimental validation

Chop Yan Lee [1,5], Dalmira Hubrich [1,5], Julia K Varga [2,5], Christian Schäfer [1], Mareen Welzel[1], Eric Schumbera[1,4], Milena Djokic[1], Joelle M Strom [1], Jonas Schönfeld [1], Johanna L Geist[1], Feyza Polat[1], Toby J Gibson [3], Claudia Isabelle Keller Valsecchi[1], Manjeet Kumar[3], Ora Schueler-Furman [2✉] & Katja Luck [1✉]

## Abstract

**Structural resolution of protein interactions enables mechanistic and functional studies as well as interpretation of disease variants. However, structural data is still missing for most protein interactions because we lack computational and experimental tools at scale. This is particularly true for interactions mediated by short linear motifs occurring in disordered regions of proteins. We find that AlphaFold-Multimer predicts with high sensitivity but limited specificity structures of domain-motif interactions when using small protein fragments as input. Sensitivity decreased substantially when using long protein fragments or full length proteins. We delineated a protein fragmentation strategy particularly suited for the prediction of domain-motif interfaces and applied it to interactions between human proteins associated with neurodevelopmental disorders. This enabled the prediction of highly confident and likely disease-related novel interfaces, which we further experimentally corroborated for FBXO23-STX1B, STX1B-VAMP2, ESRRG-PSMC5, PEX3-PEX19, PEX3-PEX16, and SNRPB-GIGYF1 providing novel molecular insights for diverse biological processes. Our work highlights exciting perspectives, but also reveals clear limitations and the need for future developments to maximize the power of Alphafold-Multimer for interface predictions.**

**Keywords** AlphaFold; Protein Interaction Interface Prediction; Linear Motifs; Benchmarking; Experimental Validation
**Subject Categories** Computational Biology; Structural Biology

## Introduction

Protein-protein interactions (PPIs) are essential for the proper functioning of essentially all cellular processes. The last decade has seen tremendous progress in the systematic mapping of human protein interactions enabling gene function prediction and the study of genotype-to-phenotype relationships (Luck et al, 2020; Drew et al, 2017; Huttlin et al, 2021). However, to understand the molecular function of individual PPIs, co-existence or mutual exclusivity of partner proteins in protein complexes, and the effect of mutations on protein function, structural information on how these proteins interact with each other is required. Unfortunately, a structure at atomic resolution is only available for ~4% of known human PPIs (Luck et al, 2020). Modular proteins interact with each other using a variety of different functional elements such as stably folded domains, intrinsically disordered polypeptide regions, short linear motifs (hereafter referred to as motifs), or coiled-coil helices forming domain-domain, domain-motif, disorder-disorder, or coiled-coil interfaces for example. Resources such as 3did (Mosca et al, 2014) or the ELM database (ELM DB) (Kumar et al, 2022) collect observed contacts between domain types and between domains and motifs, respectively. Such interface type collections can be used to predict occurrences of known interface types in protein interactions (Weatheritt et al, 2012; Mosca et al, 2013). However, it is reasonable to expect that many more protein interface types remain to be discovered. This is likely particularly true for motif-mediated PPIs, which are anticipated to number in the hundreds of thousands or millions (Tompa et al, 2014). Motifs are short stretches of amino acids in disordered regions of proteins that usually adopt a more rigid structure upon binding to folded domains in interaction partners (Davey et al, 2012). Motif-mediated interactions are of moderate binding affinity and thus, are particularly suited to mediate dynamic cell regulatory and signaling events (Van Roey et al, 2012). However, due to the transient nature of their interactions and the disorderliness of motif-containing proteins, this mode of binding is also expected to be highly understudied. Systematically generated human protein interactome maps (Luck et al, 2020; Huttlin et al, 2021) are likely a treasure trove for the discovery of novel interface types, yet no good experimental or computational methods exist to systematically map or predict protein interaction interfaces at scale.

[1]Institute of Molecular Biology (IMB) gGmbH, 55128 Mainz, Germany. [2]Department of Microbiology and Molecular Genetics, Institute for Biomedical Research Israel-Canada, Faculty of Medicine, The Hebrew University of Jerusalem, Jerusalem 9112001, Israel. [3]Structural and Computational Biology Unit, European Molecular Biology Laboratory, Heidelberg 69117, Germany. [4]Present address: Computational Biology and Data Mining Group Biozentrum I, 55128 Mainz, Germany. [5]These authors contributed equally: Chop Yan Lee, Dalmira Hubrich, Julia K Varga. ✉E-mail: ora.furman-schueler@mail.huji.ac.il; k.luck@imb-mainz.de

The release of the neural network-based software AlphaFold (AF) was not only a breakthrough for the prediction of monomeric structures of proteins (Jumper et al, 2021) but multiple studies published shortly thereafter also suggested the ability of AF to predict structures of pairwise protein interactions and complexes. Sensitivities of around 70% were reported using benchmark datasets of structurally resolved protein interactions originally developed to evaluate docking methods (Akdel et al, 2022; Bryant et al, 2022; Johansson-Åkhe et al, 2021; preprint:Evans et al, 2021). Other studies focused on structures of domain-motif interfaces to specifically evaluate AF's ability to predict structures for this mode of binding, reporting similar success rates (Akdel et al, 2022; Johansson-Åkhe et al, 2021; Tsaban et al, 2022). Only a few studies have also evaluated AF's specificity for the prediction of interface structures using controls such as random protein pairs or mutation of motifs to poly-alanine stretches (Akdel et al, 2022; Johansson-Åkhe et al, 2021; Tsaban et al, 2022). Different benchmarking studies used different versions of AF and reported on different metrics for their ability to distinguish good from bad structural models (Bryant et al, 2022; O'Reilly et al, 2023; Tsaban et al, 2022; preprint:Evans et al, 2021; Teufel et al, 2023). We generally lack a comprehensive assessment of the latest AF releases and metrics across different types of PPI interfaces for their sensitivity, specificity, and potential biases for the prediction of complex structures.

In a landmark study, researchers applied AF onto 65,000 human PPIs derived from a yeast two-hybrid-based interactome map (hereafter referred to as HuRI) and highly confident co-complex associations to structurally annotate the human interactome with AF-derived models. High confidence models were obtained for about 3000 PPIs (Burke et al, 2023). The authors noted a smaller fraction of highly confident structural models obtained for PPIs from the HuRI dataset compared to the co-complex dataset and reported that proteins in HuRI contain more intrinsic disorder and are less conserved compared to proteins from co-complex datasets. AF model confidence scores also increased for PPIs with proteins that are less disordered and more conserved, indicating that AF predictions work less well for PPIs mediated by interfaces involving disordered regions such as domain-motif interfaces, which likely dominate the human interactome (Tompa et al, 2014). However, AF benchmarking studies reported similarly high success rates for domain-motif interfaces compared to general docking benchmark datasets (Tsaban et al, 2022; Akdel et al, 2022). These discrepancies in sensitivities could be a result of two possible factors. First, they might point to differences in AF performance if small interacting fragments are used for interface prediction, as done in the benchmark studies, versus full length sequences used for structure prediction in (Burke et al, 2023). Second, these discrepancies could also point to difficulties of AF to predict structures of interface types involving disordered regions that have not been solved before, of which there are likely many in HuRI. It remains to be addressed to what extent these two possible factors contribute to the challenges encountered specifically for domain-motif interface modeling.

Determination of accuracies of novel predicted interface structures by AF ultimately requires experimentation. AF interface predictions for individual PPIs have occasionally been experimentally corroborated (Mishra et al, 2023; Bronkhorst et al, 2023). A more systematic experimental confirmation of AF interface models has been conducted using crosslinking mass spectrometry (XL-MS) (Burke et al, 2023; O'Reilly et al, 2023). While in-cell XL-MS is a very elegant approach to obtain experimental information on PPI

interfaces in unperturbed settings, it is still a method that is only accessible to few experts in the field. Other experimental approaches are needed, which can, ideally at high throughput, confirm predicted interfaces for PPIs. In this study, we thoroughly benchmarked the two most recent versions of AlphaFold-Multimer (hereafter referred to as AF) for their ability to predict domain-domain and domain-motif interfaces (DDIs and DMIs). We found that prediction accuracies drop when using longer protein fragments or full length proteins for interface predictions and developed a strategy particularly suited for the prediction of novel domain-motif interfaces in human PPIs. We applied this strategy to 62 PPIs from HuRI that connect disease-associated proteins and experimentally assessed the obtained interface predictions for seven PPIs using a plate-based bioluminescence resonance energy transfer (BRET) assay (Trepte et al, 2018) combined with site-directed mutagenesis. We identify novel interface types and report on important limitations and sources of errors in AF-derived structural models, which pave the way for future improvements in the field.

# Results

## Evaluating AlphaFold's accuracy for predicting domain-motif interfaces

To thoroughly assess the ability of AF to predict structures of binary protein complexes that are formed by a DMI, we extracted information on annotated DMI structures from the ELM DB (Kumar et al, 2022). We selected one representative structure per motif class (136 structures in total), manually defined the minimal domain and motif boundaries, and submitted the corresponding protein sequence fragments for interface prediction to AF (Fig. 1A; Dataset EV1). The domain sequences from this benchmark dataset mostly shared 20–30% sequence identity (Appendix Fig. S1A). To evaluate the accuracy of the predicted structural models, we superimposed the actual structure and predicted model on their domains and based on this superimposition, we computed the all atom RMSD between the motif of the predicted model and the actual structure (Fig. 1A). We found that 35% of the structural models were so accurately predicted that even the side chains of the motif were correctly positioned while for another 32% the backbone but not the side chains of the motif were accurately predicted. For 26% of the structures the motif was modeled into the correct pocket, but in a wrong conformation, while, for the remainder of the structures, AF failed to identify the right pocket (Fig. 1A; Dataset EV1). A similar performance was obtained when using the DockQ metric (Appendix Fig. S1B,C; Dataset EV1). This performance is unaltered when using or switching off AF's template function (Fig. S1D,E). The use of DMI structures annotated by the ELM DB enables us to explore potential differences in AF's performance regarding motif properties. We find no significant differences in average model accuracy between different categories of motif classes (two-sided Mann–Whitney test on all pairwise combinations, $n$: DEG = 10, DOC = 21, LIG = 94, TRG = 9, MOD = 2, $\alpha = 0.05$, test statistics of all pairwise combinations between 15 and 852, Appendix Fig. S1F), although the variance in model accuracy appears to differ between the motif classes. Similarly, we found no significant difference in prediction accuracy when

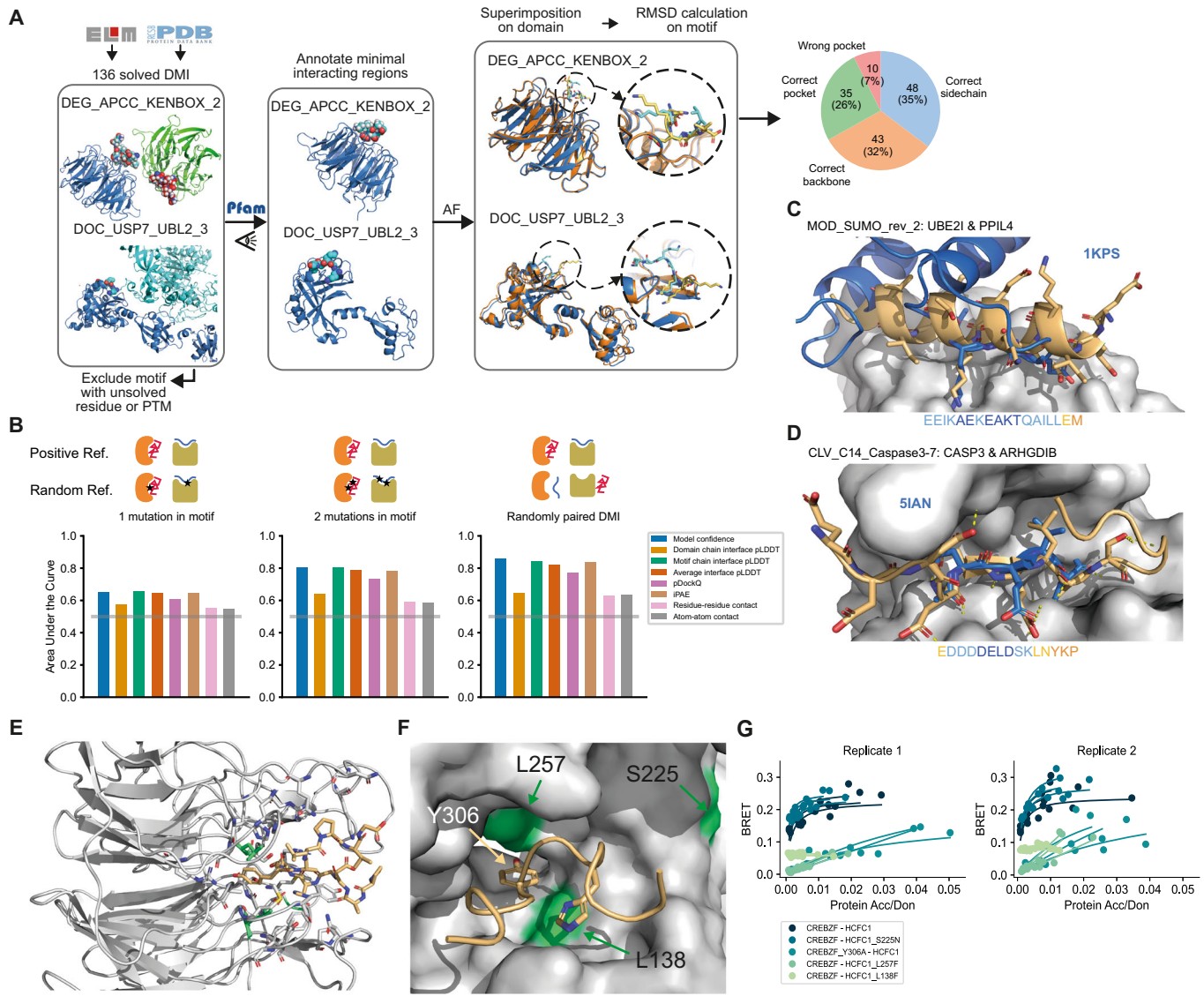

**Figure 1. Benchmarking and application of AF for DMI interface prediction using minimal interacting fragments.**

(**A**) Schematic illustrating the assembly of the DMI positive reference dataset and evaluation of AF prediction accuracies by superimposition of the solved and modeled structures. Blue and cyan indicate the domain and motif in the native structure, respectively. Orange and yellow indicate the domain and motif in the modeled structure, respectively. Proportion of structures of DMIs predicted by AF to different levels of accuracy is shown on the right. (**B**) Area under the Receiver Operating Characteristics Curve (AUROC) for different metrics using the DMI benchmark dataset as positive reference and the following different random reference sets: Left, 1 mutation introduced in conserved motif position; middle, 2 mutations introduced in conserved motif positions; right, random reshuffling of domain-motif pairs. Gray horizontal line indicates the AUROC of a random predictor. (**C**) Superimposition of AF structural model for motif class MOD_SUMO_rev_2 (orange) with homologous solved structure (PDB:1KPS) from motif class MOD_SUMO_for_1 (blue). The motif sequence used for prediction is indicated at the bottom, colored by pLDDT (dark blue=highest pLDDT). (**D**) Superimposition of AF structural model for motif class CLV_C14_Caspase3-7 (orange) with homologous structure (PDB:5IAN) solved with a peptide-like inhibitor (blue). The motif sequence used for prediction is indicated at the bottom, colored by pLDDT (dark blue=highest pLDDT). (**E**) AF prediction of a LIG_HCF-1_HBM_1 motif in CREBZF (orange) binding to the beta-propeller Kelch domain of HCFC1 (gray). Mutated domain residues for experimental testing are colored in green. (**F**) Close up on the interface shown between CREBZF and HCFC1 from (**E**). Coloring is the same as in (**E**). Key conserved motif residues are drawn as sticks. Mutated residues in the domain and motif for experimental testing are labeled. (**G**) BRET titration curves are shown for wildtype interactions and mutant constructs for CREBZF-HCFC1 pairs for two biological replicates, each with three technical replicates. Protein acceptor over protein donor expression levels are plotted on the x-axis determined from fluorescence and luminescence measurements, respectively.

stratifying by the secondary structure elements adopted by the motifs (two-sided Mann–Whitney test on all pairwise combinations, *n*: helix = 42, strand = 7, loop = 87, $\alpha = 0.05$, test statistics of all pairwise combinations between 184 and 2029, Appendix Fig. S1G), nor by how hydrophobic, symmetric, or degenerate the motif

sequence is (Pearson r < abs(0.08), $\alpha = 0.05$ Appendix Fig. S1H–J). AF models display significantly more differences to structures solved by other methods, i.e., NMR, than X-ray crystallography (two-sided Mann–Whitney test, *n*: X-ray = 115, Others = 21, $p < 0.01$, test statistics = 811, Appendix Fig. S1K) possibly because

NMR structures better represent structural dynamics that AF cannot capture, since it was trained to predict the crystallized forms of proteins.

The all-atom motif RMSD significantly anti-correlates with various AF-derived metrics (Pearson r = −0.55, p-value < 0.05 Appendix Fig. S1L,M; Dataset EV1) suggesting that these metrics are indicative of good versus bad structural models and can be used for de novo interface predictions. To evaluate AF's ability to identify high confident structural models of DMIs, we generated three different random DMI datasets. First, we randomly paired domain and motif sequences from the positive reference dataset taking into account that no motif sequence was paired with a domain sequence from the domain type that the motif is known to interact with. Second and third, we mutated one and two key motif residues, respectively, to residues of opposite chemico-physical properties. Based on the conservation of these key motif residues, we assume that the mutations would be disruptive to binding, at least when experimentally tested using minimal interacting protein fragments. Receiver operating characteristic (ROC) and precision-recall (PR) curves using the positive and random datasets (Fig. 1B; Appendix Fig. S2A,B; Dataset EV2) show that the domain interface residue pLDDT (for all metric definitions, see Methods) or the number of atoms or residues predicted to be in contact with each other, discriminated poorly between all reference datasets (AUC around 0.64). Furthermore, we observed that all tested metrics failed to discriminate interacting from non-interacting interfaces when mutating one motif residue (max AUC 0.66). However, the AF-derived metrics model confidence (preprint:Evans et al, 2021), average interface residue pLDDT, average motif interface residue pLDDT, pDockQ (Bryant et al, 2022), and iPAE (Teufel et al, 2023) discriminated well between both reference datasets when randomizing domain-motif pairs or introducing two motif mutations (max AUC 0.86, ROC statistics and ideal cutoffs can be found in Dataset EV2). We also evaluated whether the top 5 reported models by AF tend to be more similar to each other when corresponding to a correct structural model (Pozzati et al, 2022) and found that this feature has moderate predictive power (Appendix Fig. S2C).

## Application of AlphaFold for providing structural models for motif classes without available structural data

After evaluating the accuracy of AF to predict DMIs using minimal interacting regions, we aimed to use this setup for the prediction of structural models for motif classes in the ELM DB for which no structure of a complex has been solved yet. We identified 125 such motif classes based on ELM DB annotations. Of those, we selected all domain-motif instances where both the motif and the domain were derived from human or mouse proteins and submitted the corresponding domain and motif sequences for structure prediction to AF (Dataset EV3). Using a motif chain pLDDT cutoff of > 70, we obtained confident structural models for 21 motif classes. We manually inspected the structural models and noticed that even though these ELM classes have no annotations with structures, solved structures for an exact ELM instance or a very likely new instance for the ELM class are available for 11 out of the 21 cases. For most others, a close homolog structure had been solved, i.e., for LIG_MYND_3 and LIG_MYND_1, a structure solved by NMR for a LIG_MYND_2 interaction is available (Appendix Fig. S2D,E). For MOD_SUMO_rev_2, a structure of a reversed motif is available

(and annotated as such in the MOD_SUMO_for_1 class). Here it is interesting to see how very dissimilar binding modes (flexible for MOD_SUMO_for_1, helical for MOD_SUMO_rev_2), are still able to place the important binding residues in the same pockets (Fig. 1C). For CLV_C14_Caspase3-7, the structure of the caspase bound to peptide-like inhibitors has been solved (e.g. PDB:1F1J, PDB:5IAN, PDB:6KMZ), and structures of more distant caspases bound to a cleaved peptide substrate are also available. For proteases, one great advantage of AF is the ability to model both the catalytically active enzyme and an uncleaved substrate, which is practically impossible to solve experimentally (Fig. 1D).

Finally, for LIG_HCF-1_HBM_1 we were not able to identify a homologous structure in the PDB, hence, our AF-derived structural models for this motif class are likely novel. Motifs of this class are bound by the N-terminal beta-propeller Kelch domain of HCFC1 consisting of six Kelch repeats. Kelch domains have been shown to bind to motifs at a number of different sites, and thus, without prior knowledge, it is difficult to determine where the HCFC1-binding motif (HBM) would bind. HCFC1 is a transcription factor that associates with other transcription factors (Lu et al, 1997), splice factors (Ajuh et al, 2002), and cell cycle regulators (Freiman and Herr, 1997; Machida et al, 2009). We generated AF models of high confidence for the HCFC1 Kelch domain interacting with multiple motif instances that are annotated in the ELM DB. All complexes show the tyrosine of the motif docked into a deep pocket at the bottom/top of the Kelch domain (Fig. 1E,F; Appendix Fig. S2F–H), with slight variations in how the tyrosine is exactly positioned in the pocket (Fig. S2F–H). Based on clone availability we selected the structural model between HCFC1 and CREBZF for experimental validation. For this purpose, we used a BRET protein interaction assay that is based on transient overexpression of two proteins in HEK293 cells (Trepte et al, 2018). Both proteins are expressed as fusion constructs either to the Nanoluc luciferase (the donor) or mCitrine (the acceptor). Interaction of both proteins results in a BRET from the oxidized substrate of the donor to the acceptor molecule, if both are close enough to each other for the BRET to occur (see Methods for details). We observed significant binding and BRET saturation when assaying wildtype CREBZF and HCFC1 proteins (Fig. 1G; Appendix Fig. S2I,J). Mutation of the [DE]H.Y motif tyrosine to alanine (Y306A) or mutation of two residues in the Kelch domain pocket (L257F, L138F), which are modeled to be in contact with the motif tyrosine or histidine residue (Fig. 1F), strongly reduced BRET signals indicating weakening or loss of binding (Fig. 1G; Appendix Fig. S2I,J). A pathogenic mutation (S225N, source ClinVar (Henrie et al, 2018)) close to the pocket slightly reduced expression levels of HCFC1 but did not result in loss of binding (Fig. 1F,G; Appendix Fig. S2I,J). Our experiments suggest that a potential pathogenic mechanism of this mutation is not mediated via perturbed binding of partners to the Kelch repeat domain pocket of HCFC1 that we identified in this study. Unfortunately, no assertion criteria for the annotation of this mutation to be pathogenic is provided by ClinVar meaning that the mutation is either not pathogenic after all or its pathogenicity is mediated via another perturbed function not tested in this study. Collectively, these experimental results support the structural models of the HCFC1 Kelch domain pocket - motif interaction and overall provide highly confident structural models for multiple motif classes of the ELM DB without available structural information (Dataset EV4).

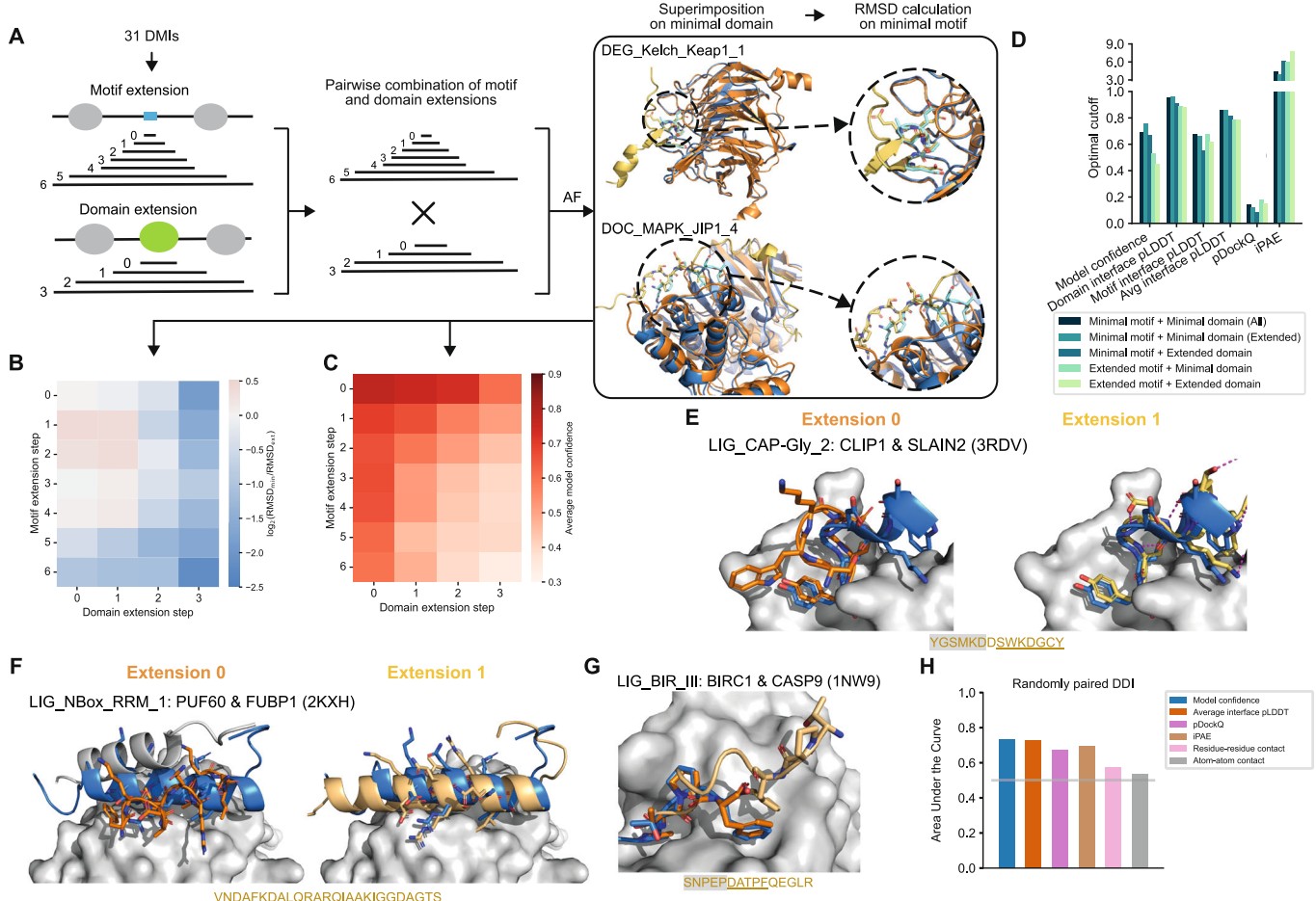

**Figure 2. Effect of protein fragment extensions on the accuracy of AF predictions.**

(A) Workflow established to assess changes in AF performance upon protein fragment extension. Blue and cyan indicate the domain and motif in the native structure, respectively. Orange and yellow indicate the domain and motif in the modeled structure, respectively. (B) Heatmap showing the fold change in motif RMSD before and after extension where positive values indicate improved predictions from extension and negative values indicate worse prediction outcomes upon extension. (C) Heatmap of the average model confidence for combinations of different motif and domain sequence extensions. (D) Optimal cutoffs derived for different metrics from ROC analysis benchmarking AF different motif and domain extensions from the reference dataset used in A and random pairings of domain and motif sequences. pLDDT-related metrics were divided by 100 for visualization purposes. (E, F) Superimposition of the structural model of the minimal (left, orange) or extended (right, yellow) motif sequence with the solved structure (motif in blue) for two different motif classes as indicated on the top of each panel. The motif sequence from the solved structure is indicated at the bottom. Motif residues are underlined, motif residues not resolved in the structure have a gray background. Sticks indicate the motif residues, domain surfaces are shown in gray based on experimental structures. (G) Superimposition of the structural model of the minimal (orange) and extended (yellow) motif sequence with the solved structure (motif in blue) for a motif instance from the motif class LIG_BIR_III. Motif sequence indicated as in (E). (H) Area under the Receiver Operating Characteristics Curve (AUROC) for different metrics using the DDI benchmark dataset as positive reference and randomly shuffled domain-domain pairs as random reference. Gray horizontal line indicates the AUROC of a random predictor.

## Evaluation of AlphaFold's ability to predict interfaces in full length proteins

Most PPIs known to date have been identified using full length protein sequences in systematic interactome mapping efforts. For the vast majority of these PPIs, no fragment or interface information is available. Thus, the question emerges how AF would perform on DMI predictions when longer protein sequences or full length proteins are submitted. To answer this question we selected 31 DMI structures from the positive reference dataset used above and generated random domain-motif pairs of those as negative control. The selected structures were sampled from different prediction accuracy categories (Fig. 1A; Dataset EV5).

We then gradually extended the motif and domain sequences by first adding flanking disordered regions, then neighboring folded domains before using the full length sequences (Fig. 2A). Comparison of the motif RMSD computed for extended versus minimal domain-motif pairs from the positive reference dataset revealed that the addition of flanking disordered regions on the motif or domain side sometimes slightly improved prediction accuracies while the addition of neighboring structured domains or the use of full length sequences led to a significant worsening of model accuracies (Fig. 2B; Dataset EV5). Interestingly, despite the fact that, for smaller extensions, model accuracies remained the same or slightly improved as determined by motif RMSD, AF-derived metrics such as the model confidence or average motif

interface residue pLDDT gradually dropped with increasing fragment length (Fig. 2C; Appendix Fig. S3A-C). ROC plots of predictions for a benchmark consisting of the positive and random domain-motif pairs revealed that, upon extension, the optimal cutoff of model confidence and iPAE considerably changed as well (Fig. 2D; Appendix Figs. S3D,E, S4A; Dataset EV6). This means that different model confidence or iPAE cutoffs are to be used depending on the length of the submitted protein sequences, which is rather impractical and thus disfavors both metrics for DMI predictions. The average motif interface residue pLDDT metric appeared to be more robust with respect to fragment length. Based on these results we chose this as the main metric and a cutoff of 70 to discriminate good from bad AF-generated DMI models regardless of fragment length.

## Extending motif sequences for interface prediction with AlphaFold reveals important motif sequence context

Various studies have highlighted that flanking sequences of motifs can influence binding affinities and specificities (Luck et al, 2012; Bugge et al, 2020). Motif annotations in the ELM DB usually refer to the core sequence of the motif, often because information on putative roles of flanking sequences is missing. In the previous section, we observed that some motif extensions notably improved AF prediction accuracies. In the hope that these cases would point to motifs with important sequence context, we manually inspected eight predictions for which the motif RMSD decreased by more than 1 Å when extending the minimal motif sequence once to the left and right by the length of the motif (extension step 1 in Fig. 2A; Appendix Fig. S4B).

By doing so interesting patterns emerged: The most prevalent contribution to increased prediction accuracies is the stabilization of the secondary structure of the motif contributed by both sidechain and backbone atoms in the flanking regions, as shown for the interaction involving the motif LIG_CAP-Gly_2 (Fig. 2E; Appendix Fig. S4C). For the LIG_NBox_RRM_1 motif, AF placed a part of the domain into the binding pocket rather than the motif, although the motif had the correct helical conformation. Elongation of the motif extended this helix, thereby increasing the interaction surface and eventually pushing out the domain's tail from the pocket (Fig. 2F). This fits with other reports where AF has been shown to predict preferential binding of competing motifs (Chang and Perez, 2023). For the LIG_HOMEOBOX class prediction, the motif is positioned in the wrong pocket unless flanking regions are included (Appendix Fig. S4C). For DOC_MAPK_JIP1_4, motif extension results in an extended motif conformation and consequently in a structural model with lower overall RMSD (Appendix Fig. S4C). For the LIG_GYF class, most models converge into an inverse orientation of the backbone except for one of the extended motifs, which lies in the binding pocket in the correct orientation (Appendix Fig. S4C). In summary, these analyses point to motif classes whose sequence boundaries could be refined.

Interestingly, for a motif instance from the LIG_BIR_III_2 class, slight motif extensions actually led to a substantial decrease in prediction accuracy. In this case, the motif is located at a neo-N-terminus that is only revealed after cleavage of the protein by a caspase (Fig. 2G). When the motif is extended in the context of the full length protein, the residues now upstream of the previous neo-N-terminus likely impede binding of the motif into the pocket due

to steric clashes. AF predicts the extended motif to bind in reversed orientation and it is mostly pushed out of the pocket. This highlights the importance of not only incorporating sequence context but also knowledge about the biological context, wherever possible, into AF modeling and model interpretation.

## Evaluating AlphaFold's performance for the prediction of domain-domain interfaces

Folded domains can not only interact with motifs but also with other folded domains forming so-called domain-domain interfaces (DDIs). To enable simultaneous prediction of DDIs and DMIs in a given protein interaction, we set out to evaluate AlphaFold's performance on DDI predictions using a reference dataset of 48 DDI structures that we manually curated out of random selections of domain-domain contact pairs extracted from 3did (Mosca et al, 2014). As a negative dataset, we randomized the pairing of these domains. Using ROC and PR statistics we found that AlphaFold performed slightly worse on this DDI benchmark dataset compared to its performance on DMIs (max AUC 0.73 vs. 0.86) (Fig. 2H; Appendix Fig. S4D–F; Dataset EV7) but still showed significant discriminative power. Interestingly, the best performing metric for DDI predictions was the average interface pLDDT score with an optimal cutoff of 75, which ranked fourth for DMI predictions.

## Comparison of AlphaFold v2.2 with v2.3

During the course of our work, AF multimer version 2.3 was released. To determine whether the new release improved DMI and DDI prediction accuracies, we repeated all benchmarking with AF v2.3 and found that motif RMSDs and other AF-derived metrics on average improved compared to AF v2.2 when using minimal interacting fragments (Appendix Fig. S5A–D; Dataset EV1, two-sided Wilcoxon signed-rank test on motif all atom RMSD: $n = 136$, $W = 2413$, $p < 0.0001$). AF v2.3 still showed a decrease in prediction accuracy when using extended protein fragments but this decrease was less pronounced compared to the corresponding decrease for v2.2 (Appendix Fig. S5E,F; Dataset EV5). Despite these improvements on the sensitivity side of AF, when benchmarked against random datasets, overall prediction accuracies only slightly improved compared to v2.2 (Appendix Fig. S5G,H; Appendix Fig. S6A–C; Dataset EV2, EV6, EV7, EV8).

## Application of AlphaFold for the discovery of novel interfaces in protein interactions without any a priori interface information

Since the use of larger or full length protein sequences leads to a poor sensitivity for DMI predictions by AF, we devised the following strategy for the use of AF for interface predictions for known protein interactions: Using AF models of the full length monomeric structures of both interacting proteins, we decided on boundaries between structured domains and disordered regions based on manual inspection (see Methods). We then fragmented the disordered regions by designing overlapping fragments varying in length from ten residues up to the length of the respective disordered region (Fig. 3A). We then paired disordered with ordered, and ordered with ordered fragments for interface prediction by AF (Fig. 3A). To assess to which extent this

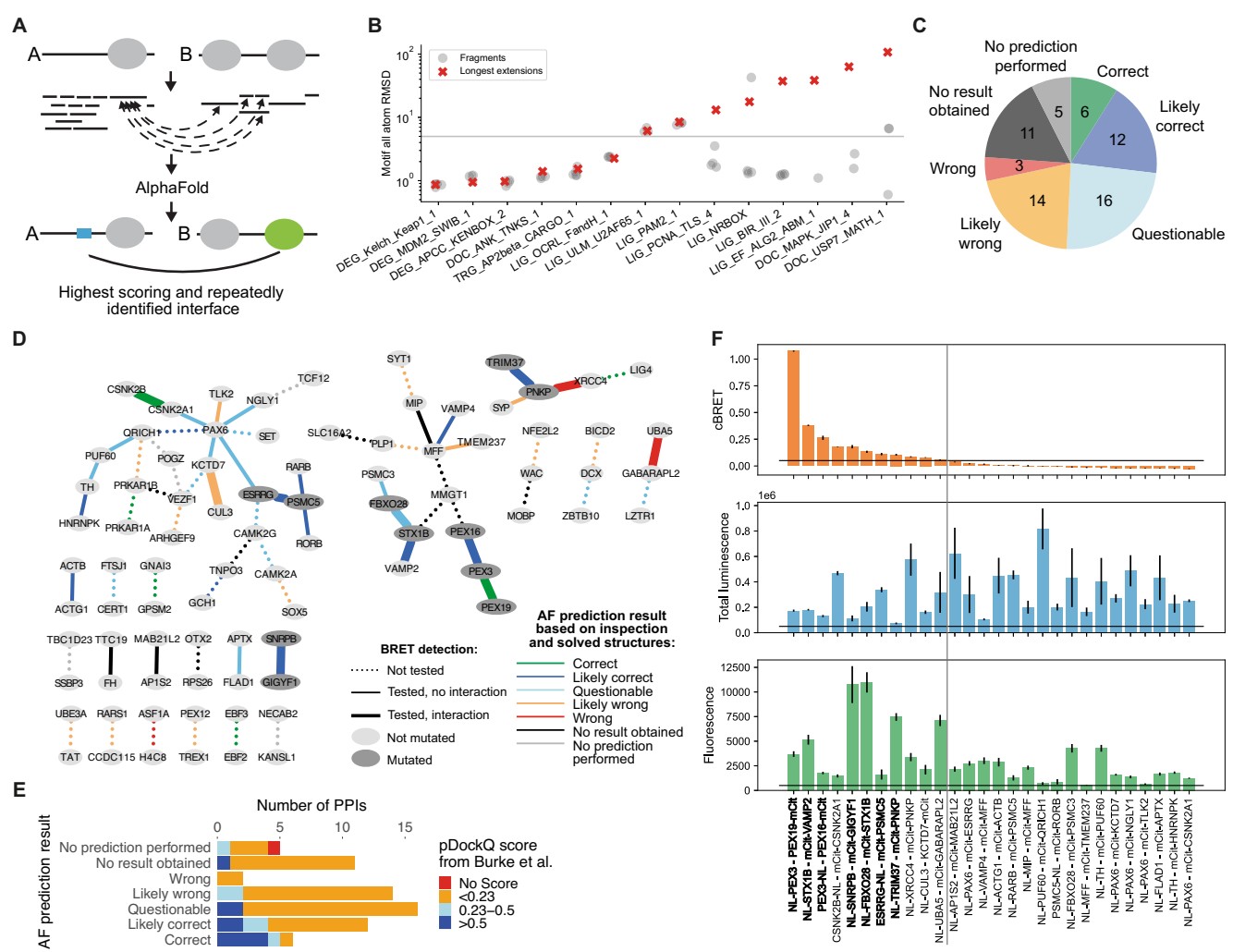

**Figure 3.   AF prediction and experiments on PPIs connecting NDD proteins.**

(**A**) Schematic of the fragmentation approach applied on a pair of interacting proteins, A and B. Proteins are fragmented into folded and disordered regions based on manual inspection. Disordered regions are further fragmented. All disordered and folded fragments of one protein are paired with the folded regions of the other protein and vice versa for AF prediction. (**B**) Accuracy measured in motif RMSD compared to native structures for models obtained from fragmenting proteins from 20 DMIs from the positive reference dataset and comparison to model accuracy obtained when using (near) full length proteins for structure prediction (red crosses). Only models that meet the cutoff for identifying high confident models are shown. Six DMIs did not result in any such model. The gray horizontal line indicates the RMSD cutoff used to identify accurate models (see methods for details). (**C**) AF prediction outcome on 67 HuRI PPIs connecting NDD proteins. (**D**) PPI networks illustrating AF prediction outcomes and experimental retesting of PPIs in BRET assay. (**E**) Number of PPIs connecting NDD proteins with structural models at indicated pDockQ cutoffs from (Burke et al, 2023) grouped based on AF prediction outcomes using the fragmentation approach as shown in (**C**). (**F**) cBRET, total luminescence, and fluorescence for 28 PPIs connecting NDD proteins that were tested in the BRET assay. Luminescence and fluorescence measurements indicate expression levels of NL and mCit fusion proteins, respectively. Black horizontal lines indicate expression level and PPI detection cutoffs. The gray vertical line separates the detected (left) from undetected PPIs. Protein pairs in bold indicate those selected for interface validation via site-directed mutagenesis. Error bars indicate STD of three technical replicates. Source data are available online for this figure.

fragmentation approach would lead to an increase in sensitivity but also in false model predictions, we selected 20 out of the 31 DMI structures that were previously used to investigate the effect of fragment extension on prediction accuracies. We attempted model prediction with the full length sequences of these 20 DMI pairs and obtained a model for two of which only one met the motif interface pLDDT cutoff and corresponded to an accurate prediction (TRG_AP2beta_CARGO_1 in Fig. 3B; Dataset EV9, see methods for details). We then switched to using fragment extension step 5 for motifs and/or 2 for domains (Fig. 2A) and obtained accurate

models for an additional 5 of the 20 DMI pairs. Applying the full fragmentation approach onto all 20 DMI pairs resulted in accurate model prediction for an additional 6 DMI pairs (Fig. 3B) representing an increase in sensitivity for full length vs fragments from 5 to 60%. We then shuffled the 20 DMI pairs to generate 20 random DMI pairs for which we performed the fragmentation approach. As expected from an earlier estimated 20% false positive rate (FPR) (Appendix Fig. S4A), 19 of the 20 random protein pairs had at least one fragment pair that produced a model above the motif interface pLDDT cutoff (Appendix Fig. S6D; Dataset EV9)

indicating that predictions done using this fragmentation approach can substantially increase sensitivity while also producing a considerable number of false models using the established scoring metrics. This needs to be taken into account when modeling new interactions with this fragmentation strategy, as covered in the following section.

We selected PPIs from HuRI that connect proteins associated with neurodevelopmental disorders (NDDs) and subjected these to our AF fragmentation pipeline to predict putative DMIs and DDIs. For 51 out of 62 PPIs we obtained at least one structural model of significant confidence (Fig. 3C,D). In retrospect, manual inspection of the predictions obtained for these PPIs revealed that, for 9 PPIs, a solved structure of the interface was already available. Reassuringly, six out of these were accurately predicted by AF. For the remainder of the PPIs, 12, 16, and 14 resulted in a likely correct, questionable, or likely wrong prediction, respectively, based on manual inspection of the models (Fig. 3C,D; Dataset EV10). Likely wrong predictions were scored as such based on docking of the protein partner into nucleic acid or metal ion binding or catalytically active sites. We also considered structural models as likely wrong, if different protein fragments of the partner were predicted with similarly high scores to bind to the same pocket on the domain. More detailed information can be found in Methods and Appendix Text S1. Of note, for 8 of the 12 PPIs with a likely correct prediction, AF predictions performed using the full length proteins (Burke et al, 2023) did not result in a high confidence prediction (Fig. 3E). 28 of the 62 PPIs were in our hands amenable to experimental testing using the BRET assay introduced earlier (see Methods for details). Significant BRET signals were observed for 11 of these 28 PPIs (Fig. 3F). Of those, 7 PPIs were selected for validating the predicted interfaces (Fig. 3D,F). The remaining four PPIs were not further considered because for three of them a structure already exists (CSNK2B-CSNK2A1, PNKP-XRCC4, UBA5-GABRAPL2) and for the fourth interaction (KCTD7-CUL3) we classified the predicted interface as likely wrong. Next, we will first describe failures in validating predicted interfaces followed by the successes.

For the interaction between PNKP and TRIM37, we obtained high confident structural models involving two different interfaces. AF predicted the PNKP FHA domain to bind to several disordered stretches in TRIM37 (Fig. 4A) that are overall negatively charged. These short regions were predicted to bind to a pocket on the FHA domain that is known to bind phosphorylated threonines (Durocher et al, 2000), which led us to conclude that these predictions were likely wrong. AF also predicted the MATH domain of TRIM37 to bind to two separate disordered putative motifs located between the FHA domain and phosphatase domain in PNKP (Fig. 4A–C). However, none of the mutants aimed at disrupting the predicted interfaces (Fig. 4B) involving the MATH domain showed a decrease in BRET signal compared to wildtype (Fig. 4D; Appendix Fig. S7A) indicating that TRIM37 and PNKP do not interact with each other via this interface.

AF predicted with high confidence binding of PSMC5 to the hormone receptor domain of ESRRG via two distinct motifs (Fig. 4E–G) with similarity to LxxLL motifs known to bind this type of domain (LIG_NRBOX in ELM DB). We reproducibly found that none of the motif mutations in PSMC5 decreased binding to ESRRG compared to wildtype while both domain pocket mutations led to a remarkable reduction in BRET signal (Fig. 4H; Appendix

Fig. S7B,C) indicating that PSMC5 might bind to ESRRG via this pocket but not with the predicted motifs.

AF predicted a coiled-coil interface between STX1B and VAMP2 of moderate confidence (Fig. 5A,B). STX1B is a close homolog to STX1A, which binds in a 4-helix bundle to VAMP2 together with SNAP25 in a 1:1:2 stoichiometry, respectively, as observed by crystallography (PDB:1N7S (Ernst and Brunger, 2003)). This structure together with our predictions suggest that STX1B might bind VAMP2 in a similar way. Indeed, removal of the single helical SNARE domain in STX1B led to complete loss of binding to VAMP2 (Fig. 5C; Appendix Fig. S8A,B). Interestingly, FBXO28 was predicted by AF to bind to STX1B via a similar coiled-coil interface involving an extended helix in FBXO28 and the SNARE domain in STX1B (Fig. 5A,D). Here, deletion of the SNARE domain in STX1B or of the extended helix in FBXO28 reproducibly reduced, but did not abolish the interaction between STX1B and FBXO28 (Fig. 5E; Appendix Fig. S8C,D). We identified three pathogenic or likely pathogenic mutations in the SNARE domain of STX1B in ClinVar of which V216E and G226R are associated with generalized epilepsy with febrile seizures plus, type 9. Testing all three mutations in the BRET assay we observed a drastic decrease in binding for STX1B V216E to FBXO28 (Fig. 5F; Appendix Fig. S8C,D). However, the measured effects of the mutations on the FBXO28-STX1B interaction do not correlate with their location at the predicted interface. V216E, for example, is not predicted to be in contact with residues of FBXO28 (Fig. 5D). This indicates that the actual predicted orientation of the two extended helices with respect to each other is likely incorrect.

The fact that the deletion of the extended helix in FBXO28 or the SNARE domain in STX1B reduced but did not abrogate binding of both proteins to each other (Fig. 5E) suggests that a secondary interface might exist. Indeed, AF predicted additional interfaces between FBXO28 and STX1B involving folded and disordered regions in both proteins (interfaces i and ii in Fig. 5A). Mutations designed to disrupt these interfaces partially confirmed the involvement of some of these regions in binding as assayed with BRET (Appendix Fig. S8E–H). In addition, the pathogenic mutation R348L in FBXO28 predicted to be at interface ii seemed to increase binding to STX1B (Appendix Fig. S8I–L). In summary, our experimental data indicate that multiple regions of FBXO28 and STX1B may be involved in the binding but the exact structural details of this interaction remain to be elucidated. In the following two sections, we will describe in more detail successful interface validations for interactions involving PEX3, PEX19, and PEX16 as well as SNRPB and GIGYF1.

## PEX3, PEX19, and PEX16

The interaction interface between PEX19 and PEX3 has been structurally resolved before and consists of an interaction between an N-terminal motif in PEX19 that binds to the cytosolic alpha-helical domain of PEX3 (PDB:3MK4, (Schmidt et al, 2010)). Using corresponding protein fragments, AF predicted a structural model that is highly similar to the solved structure (Fig. 5G; Appendix Fig. S9A,B). We introduced mutations in the PEX19 motif and PEX3 pocket (Appendix Fig. S9A) and found that F29K in the motif weakened but clearly maintained BRET binding signals indicating the existence of a secondary binding site between both proteins (Fig. 5H; Appendix Fig. S9C,D). Indeed, AF predictions with other

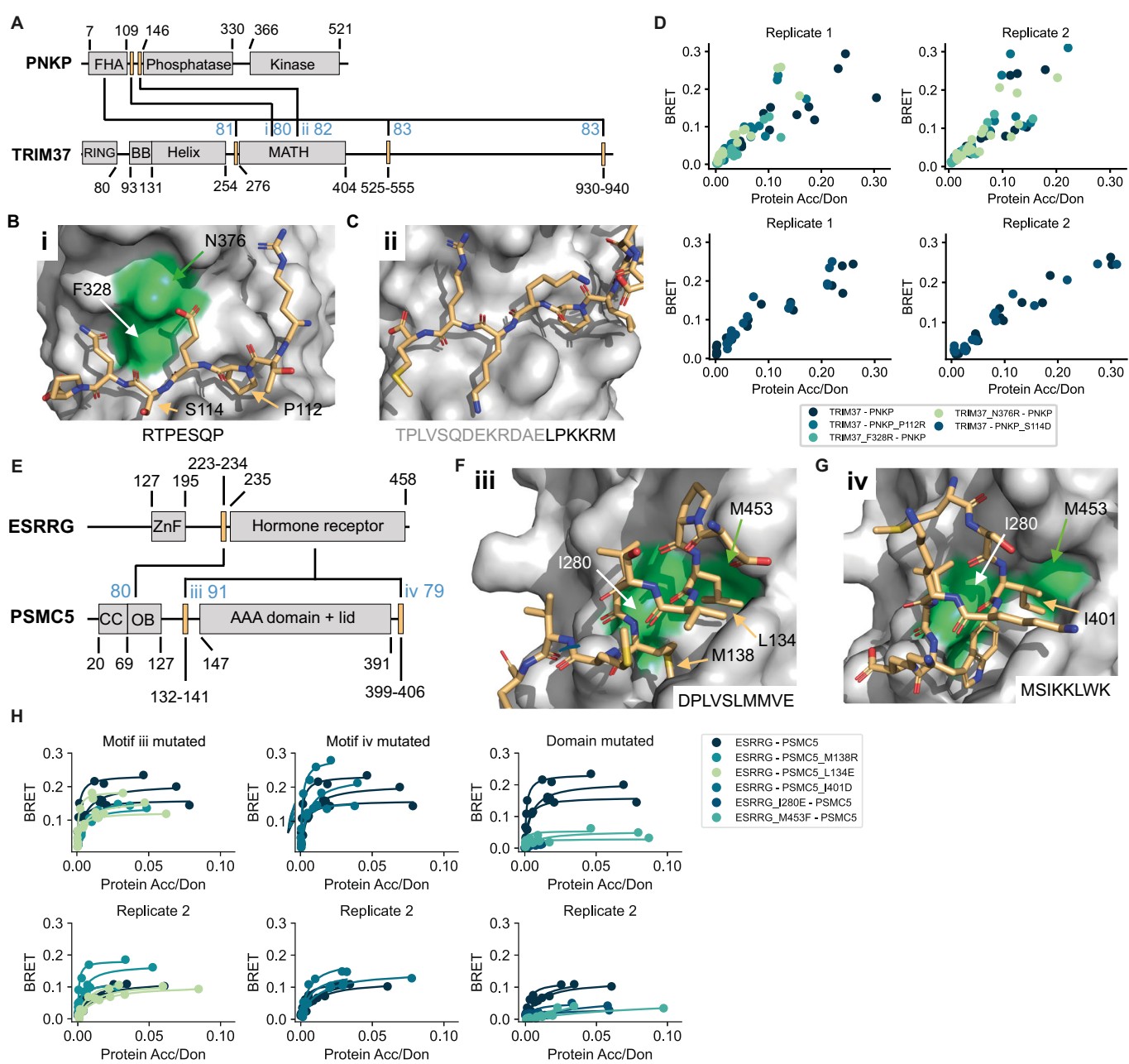

**Figure 4. Verification of interface predictions for TRIM37-PNKP and ESRRG-PSMC5.**

(**A**) Schematic of the domain architecture of PNKP and TRIM37 with indication of top predicted interfaces. Numbers in blue indicate the motif interface pLDDT for the respective interface. Roman numbering refers to structural models in (**B**) and (**C**). (**B**) Structural model of interface i shown in (**A**) with labeled residues that were mutated. (**C**) Structural model of interface ii shown in (**A**). (**D**) BRET titration curves are shown for wildtype interaction and mutants for two biological replicates, each with three technical replicates. Protein acceptor over protein donor expression levels are plotted on the *x*-axis determined from fluorescence and luminescence measurements, respectively. The BRET trajectory could not be fitted because of an unusual saturation behavior (see methods for details). (**E**) Schematic of the domain architecture of ESRRG and PSMC5 with indication of top predicted interfaces. Numbers in blue indicate the motif interface pLDDT for the respective interface. Roman numbering refers to structural models in (**F**) and (**G**). (**F**) Structural model of interface iii shown in (**E**) with labeled residues that were mutated. (**G**) Structural model of interface iv shown in (**E**). (**H**) BRET titration curves are shown for wildtype interaction and mutants of ESRRG-PSMC5 pairs for two biological replicates, each with three technical replicates. Protein acceptor over protein donor expression levels are plotted on the *x*-axis determined from fluorescence and luminescence measurements, respectively. In panels (**B**), (**C**), (**F**), and (**G**) motif sequences are indicated at the bottom. Gray letters indicate residues not predicted to bind. Source data are available online for this figure.

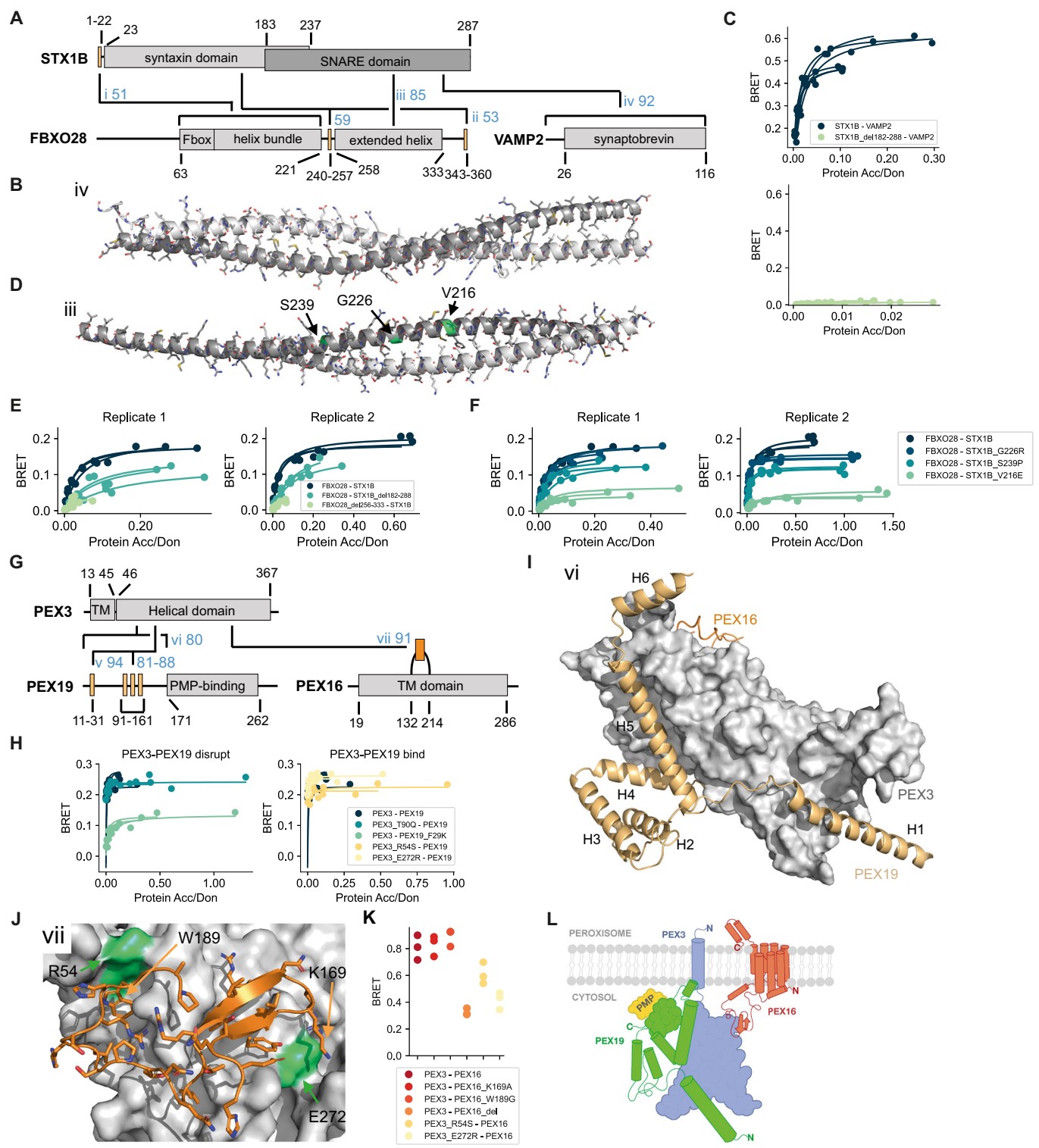

disordered fragments of PEX19 paired with the PEX3 domain resulted in highly confident models for interfaces involving a binding pocket on PEX3 that is distal to the pocket where the N-terminal PEX19 motif is known to bind. When using a protein fragment that spans the full disordered N-terminal region of PEX19 (1–170), AF predicts the known PEX3-binding motif and helix 4

and 5 to dock into the primary and secondary pocket, respectively (Fig. 5G,I), supporting simultaneous interaction via both interfaces.

While the interaction between PEX3 and PEX16 has been described before, little is known about how both proteins interact with each other. The monomeric AF model of PEX16 shows a helical fold, which could in its entirety be transmembrane (TM).

**Figure 5. Verification of interface predictions for STX1B-FBXO28, STX1B-VAMP2, PEX3-PEX19, and PEX3-PEX16.**

(A) Schematic of the domain architecture of STX1B, FBXO28, and VAMP2 with indication of top predicted interfaces. Numbers in blue indicate the motif interface pLDDT (for order-disorder fragment pairs) or average interface pLDDT (for ordered-ordered fragment pairs) for the respective interface. Roman numbering refers to structural models in (B), (D), Appendix Fig. S8E, and Appendix Fig. S8I. (B) Structural model of interface iv shown in (A). In panel (B) and (D), the chains are color-coded according to the colors of the domains in (A). (C) BRET titration curves are shown for wildtype interactions and deletion constructs for two biological replicates, each with three technical replicates. Protein acceptor over protein donor expression levels are plotted on the x-axis determined from fluorescence and luminescence measurements, respectively. (D) Structural model of interface iii shown in (A) with tested pathogenic mutations labeled and colored in green. (E, F) BRET titration curves are shown for wildtype interactions and deletion constructs for two biological replicates, each with three technical replicates. Protein acceptor over protein donor expression levels are plotted on the x-axis determined from fluorescence and luminescence measurements, respectively. (G) Schematic of the domain architecture of PEX3, PEX19, and PEX16 with indication of top predicted interfaces. Numbers in blue indicate the motif interface pLDDT for the respective interface. Roman numbering refers to structural models in (I), (J), and Appendix Fig. S9A. Region vi covers residues 1–170, which includes the previously reported N-terminal motif as well as three putative motifs suggested by the AF models. (H) BRET titration curves are shown for wildtype interaction and mutants of PEX3-PEX19 pairs for three technical replicates. Protein acceptor over protein donor expression levels are plotted on the x-axis determined from fluorescence and luminescence measurements, respectively. The left plot displays mutants aimed at disrupting binding between PEX3-PEX19 while the right plot displays mutants aimed at disrupting the PEX3-PEX16 PPI why binding between PEX3-PEX19 should not be altered. (I) Superimposition of structural models of interface vi (PEX3-PEX19) and vii (PEX3-PEX16) on the PEX3 domain. Note that modeling smaller fragments of PEX19 generates alternative interactions with the binding sites. (J) Structural model of interface vii shown in (G). (K) BRET values with subtracted bleedthrough for PEX3-PEX16 wildtype and various mutated constructs. Three technical replicates are shown. (L) Proposed model for how the trimeric complex of PEX3, PEX19, and PEX16 might assemble at the peroxisomal membrane. Source data are available online for this figure.

Between the putative TM helix 4 and 5 there is a large loop (132–214), which was predicted by AF with very high confidence to bind to a third pocket on the PEX3 domain, opposite to both binding sites mentioned earlier for PEX19 (Fig. 5G,I,J). Of note, different fragments of this loop as well as the entire PEX16 were repeatedly predicted to bind in similar modes to PEX3, further increasing the confidence in this prediction. Encouraged by these results, we submitted all three full length PEX sequences for complex prediction to AF and obtained a model that supports simultaneous binding of PEX16 and PEX19 to PEX3 (Appendix Fig. S9E). We individually mutated two residues in the PEX16 loop, deleted the loop in its entirety (del162-192), and mutated two residues on PEX3 (highlighted in Fig. 5J). Unfortunately, higher expression levels of PEX16 seem to trigger degradation of PEX3 (Appendix Fig. S9F), which we did not observe for the same constructs when co-expressed with PEX19 (Appendix Fig. S9G). As a consequence, we could not obtain titration curves and BRET50 estimates but obtained reliable BRET signals for lower PEX3-PEX16 DNA transfection ratios showing that the deletion as well as both PEX3 mutants significantly decreased binding to PEX16 (Fig. 5K; Appendix Fig. S9H). Of note, these PEX3 mutants (R54S and E272R) did not alter binding to PEX19, showing that the overall structural integrity of PEX3 was not perturbed by these mutations (Fig. 5H; Appendix Fig. S9D).

PEX3 and PEX19 are peroxin proteins that regulate peroxisome homeostasis. PEX16 is believed to serve as an integral membrane-bound receptor for PEX3 (Matsuzaki and Fujiki, 2008) while PEX3 is thought to serve as a docking site for PEX19 (Fujiki et al, 2006). PEX19 in turn is a cytosolic carrier for peroxisomal membrane proteins to the peroxisome (Fujiki et al, 2006). Combining results from previously published functional studies with the structural and experimental results obtained in this study, a model for a trimeric complex between PEX3, PEX19, and PEX16 emerges (Fig. 5L) where PEX16 fully inserts into the peroxisome membrane via a fold that consists of seven helices (residues 19-286) with its N-terminal end being cytosolic and its C-terminal end protruding into the peroxisome. The extended loop between TM helix 4 and 5 reaches into the cytosol and docks onto PEX3, which is further anchored into the peroxisomal membrane via its N-terminal TM helix (residues 13–45). PEX19 docks onto PEX3, opposite to where PEX16 is bound, via two interaction surfaces—one corresponding

to the known PEX3-binding motif in PEX19 and a second one corresponding to a novel motif (residues 99–146) docking at a hitherto unknown second binding site on PEX3 for PEX19. This model explains how PEX3 is anchored to the peroxisomal membrane via PEX16 and how PEX3 can bind very tightly PEX19, which can then deliver PMPs to the peroxisome. Mutations in any of the three PEX proteins are associated with severe developmental phenotypes referred to as peroxisome biogenesis disorders (Fujiki et al, 2022). The vast majority of the around 150 mutations annotated for the three proteins are uncharacterized (Henrie et al, 2018), dozens of which fall into the predicted interfaces. The structural models obtained from this work can inform future studies aimed at characterizing the effects of these mutations.

## SNRPB and GIGYF1

AF predicted two different types of interfaces with high confidence for the interaction between SNRPB and GIGYF1. The first interface involves the LSM domain of SNRPB which was predicted to bind to various fragments in the long disordered regions of GIGYF1 (Fig. 6A). These regions do not display any common sequence pattern. The structure of SNRPB has been resolved as part of the Sm ring complex that binds small nuclear RNA (PDB:4WZJ, (Leung et al, 2011)) showing that the surface on the LSM domain predicted to bind to disordered fragments of GIGYF1, is actually engaged in binding LSM domains of other Sm proteins within the complex (Fig. 6B). We thus conclude that these predictions are likely wrong. The second type of interface predicted by AF involves the GYF domain in GIGYF1 and multiple short disordered fragments in the C-terminal region of SNRPB, which repeatedly carry the sequence PPPGM(R) (Fig. 6A,C). We designed various deletion constructs of SNRPB that would gradually remove more and more of the repeated proline-rich motif. We observed, using the BRET assay, that these deletion constructs gradually decreased binding to GIGYF1 (Fig. 6D; Appendix Fig. S10A,B). We also mutated the GYF domain pocket and found that W498E but not L508F would decrease binding to SNRPB (Fig. 6D,E; Appendix Fig. S10A–D). To further corroborate these findings we performed a co-immunoprecipitation experiment, where endogenous GIGYF1 interacted with HA-tagged full length SNRPB (Fig. 6F). This

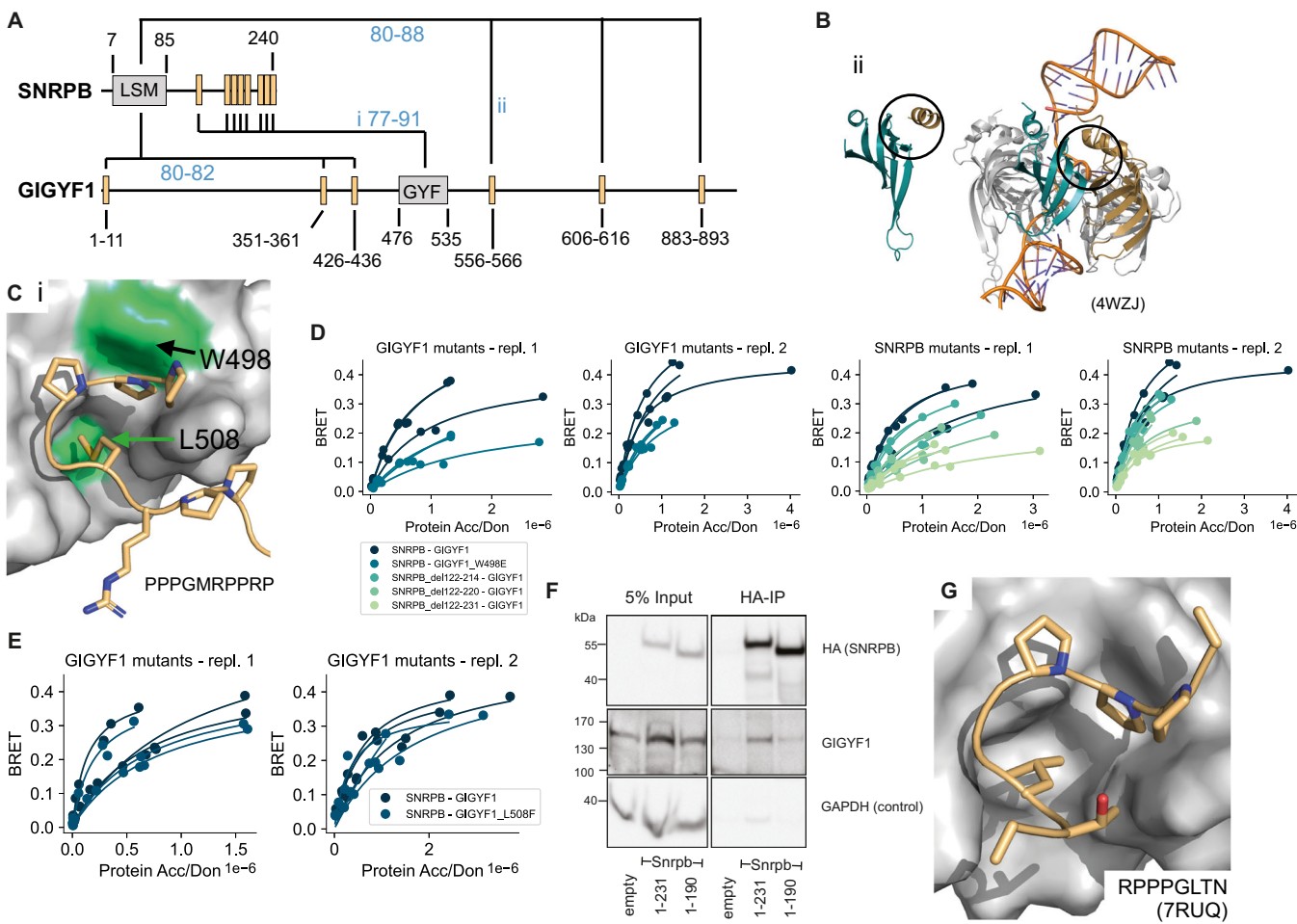

**Figure 6. Verification of interface predictions for SNRPB-GIGYF1.**

(A) Schematic of the domain architecture of SNRPB and GIGYF1 with indication of top predicted interfaces. Numbers in blue indicate the motif interface pLDDT for the respective interface. Roman numbering refers to structural models in (B) and (C). (B) Structural model of interface ii shown in (A) (left) and in comparison a solved structure (PDB:4WZJ) of the Sm ring complex (right) bound to RNA (orange). The LSM domain of SNRPB is shown in cyan. The position of the predicted motif (left) or neighboring LSM domain of SNRPD3 (right) are indicated in gold. Black circles indicate the predicted interface in the model and corresponding interface in the complex on the LSM domain of SNRPB. (C) Structural model of interface i shown in (A) with tested domain mutations labeled and colored green. The motif sequence is indicated at the bottom. (D, E) BRET titration curves are shown for wildtype interactions, deletion constructs of SNRPB, and single point mutants in GIGYF1 for two biological replicates, each with three technical replicates. Protein acceptor over protein donor expression levels are plotted on the x-axis determined from fluorescence and luminescence measurements, respectively. (F) Cropped immunoblot of input (5%) and HA antibody immunoprecipitation (IP) performed in parental HEK cells (empty, untagged negative control), Snrpb(full-length, 1-231)-2xHA-mNeonGreen, Snrpb(1-190)-2xHA-mNeonGreen expressed from a single locus in Flp-In™ T-REx™ 293 Cell Lines. The HA antibody was used for detecting the immunoprecipitated Snrpb-proteins, endogenous GIGYF1 was detected with GIGYF1 antibody, GAPDH serves as a loading and negative-IP control. The experiment was performed twice with equivalent outcome, one representative experiment is shown. (G) Solved structure (PDB:7RUQ) of the GYF domain of GIGYF1 bound to a proline-rich motif in TNRC6C. The sequence of the motif in TNRC6C is indicated. Source data are available online for this figure.

interaction appeared less pronounced upon truncation of the C-terminal proline-containing region of SNRPB (Fig. 6F). This further suggests that both proteins interact with each other in cells and that this interaction is stabilized by the predicted interface.

During the course of these studies, a structure was published (PDB:7RUQ, Sobti et al, 2023) showing binding of the GYF domain of GIGYF1 to a motif of sequence PPPGL of the protein TNRC6C confirming the binding mode predicted by AF where a hydrophobic residue (M or L) inserts into a hydrophobic pocket and where the proline residues contact the surrounding domain surface (Fig. 6C,G). Interestingly, this hydrophobic pocket does not exist in the previously solved structure of the GYF domain of CDBP2 binding to a proline-rich peptide that is flanked by positively

charged residues establishing important contacts with the domain (PDB:1L2Z, (Freund et al, 2002)). This structure formed the basis for the definition of the LIG_GYF motif class in the ELM DB. The recently resolved structure of the GYF domain of GIGYF1 together with our structural models and experimental validations argue for an extension of the existing motif definition or definition of a new motif subclass.

## Discussion

AF has revolutionized the field of structural bioinformatics and has sparked much excitement about its potential to predict structures of

interacting proteins and bringing us closer to a structurally resolved protein interactome. However, from existing studies it largely remained unclear whether AF's performance depends on the type of interfaces and the length of submitted protein chains for interface prediction, which metrics perform best in identifying likely correct structural models of interfaces, how specific AF predictions are, and to which extent highly confident structural models can be experimentally corroborated. In this study, we showed that AF performs similarly well for interfaces between folded domains and interfaces formed between a folded domain and a short linear motif. Using minimal interacting regions for interface prediction we reached sensitivities of up to 80% similar to previously published work (Tsaban et al, 2022; Johansson-Åkhe et al, 2021). We thoroughly investigated AF's FPR using random domain-motif pairs and found it to be around 20%. However, asking AF to discriminate binders from non-binders when motif sequences carried one disruptive mutation, we found that prediction accuracies were close to random. This points to an important limitation in AF's ability to predict binding specificities and is in line with previous reports on AF's inability to predict the effect of mutations (Buel and Walters, 2022). Comparison of different metrics to discriminate good from bad structural models using either minimal interacting fragments or extensions revealed the average interface pLDDT for DDI models and the motif interface pLDDT for DMI models to be the most robust and best performing metrics. However, when manually inspecting AF predictions we found it useful to also consider AF's model confidence, suggesting that in the future a combination of different metrics might be even more powerful to discriminate good from bad structural models. The alignment depth has been previously reported to somewhat influence model accuracy (Bryant et al, 2022). While this feature was not investigated here, it might serve as a pre-filter to identify PPIs of high conservation for which structural modeling will likely be more successful. Interestingly, the number of residues or atoms predicted to be in contact with each other was poorly predictive, in contrast to a previous report (Bryant et al, 2022), confirming our observations that the tested AF versions in this study will always put both chains in contact with each other to create atomic contacts, and from visual inspection alone it is very challenging to tell good from bad structural models apart. Of note, observed differences in AF performance across studies likely originate both from using different benchmark datasets and different AF versions. Our study is unique in that it assesses multiple metrics on two different classes of interfaces, DMIs and DDIs, using two different AF versions. More work is needed to develop benchmark datasets of coiled-coil and disorder-disorder interfaces to also evaluate AF's performance for these modes of binding. Of note, our benchmark datasets almost exclusively consisted of structures that AF has seen in the training process. Interestingly, benchmark studies done with unseen structures reported similar sensitivities (preprint:Bret et al, 2023) indicating that AF is not strongly biased towards structures it has seen before.

We extensively explored the influence of protein fragment length on AF's performance and found that slight extensions of minimal motif sequences can improve prediction accuracies. Inspection of individual cases revealed novel information on important motif sequence context that was so far missing in corresponding motif entries at the ELM DB. However, longer disordered fragments or fragments containing ordered and large disordered regions generally decrease AF prediction accuracies as also reported in a recent preprint (preprint:Bret et al, 2023). Furthermore, optimal cutoffs for various metrics such as the model confidence decreased when using longer protein fragments, making them less robust for interface prediction with AF. When evaluating performance differences for longer and shorter protein fragments we identified three DMI pairs involving the motif classes DEG_APCC_KENBOX_2, LIG_Pex14_3, and LIG_GYF, for which, during fragment extension, a second known motif occurrence was added to the fragment. This second motif was selected by AF during interface prediction, displacing the original motif and leading to a high RMSD score. We removed these instances from the dataset when evaluating AF's performance on fragment extension but they point to biologically correct variability in AF prediction outcomes due to existing multivalency of many DMIs in protein interactions. Other work suggested that AF is able to select the stronger binder among two motif occurrences (Chang and Perez, 2023), which might at least in some cases guide AF motif selections. However, in other cases this motif preference might also hinder discovery of multivalency in PPIs. For example, the use of smaller protein fragments for the protein pair SNRPB and GIGYF1 enabled the discovery of a proline-rich repeat motif in SNRPB.

In comparison to predictions made using full length proteins (Burke et al, 2023) we found that protein fragmentation increased the probability of obtaining a high confidence interface prediction, especially for cases involving proteins with long disordered regions such as GIGYF1. For smaller and more globular proteins like the PEX proteins studied above, full length predictions can identify the right binding sites but these can be further substantiated by running additional predictions with smaller fragments. The fragmentation approach increases the number of prediction runs per protein pair from one to a couple hundred, depending on the length and modularity of both proteins. The vast majority of these fragment pairs should not interact. With a FPR of 20%, this means that more actual non-interacting than truly interacting fragment pairs will result in a high confidence prediction. A big challenge is thus to identify likely correct interface predictions among the many false ones. This is also illustrated by the prediction results that we obtained for the seven protein pairs that we followed up experimentally. Clearly, AF's general limited specificity contributes to these false predictions. We observed that additional sources of error can arise from exposed intramolecular binding sites resulting from fragmentation, incorrectly designed boundaries of folded regions, and docking of protein fragments into enzymatic pockets of metabolic enzymes or sites for metal ion, DNA, or RNA binding. It seems that AF is overall well suited to find binding pockets on folded domains. However, our work also clearly demonstrates that AF is able to correctly dock the matching partner structure into these pockets without the need for a pre-existence of both partner structures in the bound conformation contrary to other state-of-the-art docking algorithms. AF's high sensitivity with respect to intramolecular binding sites and wrongly fragmented folded regions will make it particularly hard to fully automate the fragment design process. Despite these challenges we found that recurrent interface predictions from overlapping fragments can help gain confidence in predictions, as also highlighted in a recent study (Bronkhorst et al, 2023), since we rarely observed this recurrence for likely wrong predictions.

Given the reported uncertainties in AF predictions, even for high confidence cutoffs, experimental validation is essential. The BRET assay used here has been shown in previous studies to be sensitive enough to quantify weakening of binding introduced by point mutations and to detect motif-mediated PPIs (Ebersberger et al, 2023; Trepte et al, 2018; Mo et al, 2022). Using the BRET assay, we were able to detect 11 out of 28 PPIs from the HuRI dataset. This retest rate is actually higher compared to retest rates of gold standard PPI datasets used in the past to benchmark various binary PPI assays including this BRET assay, attesting the overall detectability of PPIs from HuRI (Braun et al, 2009; Trepte et al, 2018; Choi et al, 2019). The NL and mCit fusions used in the BRET assay allowed us to monitor the expression levels of wildtype and mutant constructs, which is important to rule out loss of binding because of a destabilization of the protein. However, we cannot exclude the possibility that some expressed mutants might still be partially unfolded or mislocalized and thus, some loss of binding detected in our study could be unspecific and not the result of a specific perturbation of the predicted interface. Furthermore, preservation of binding observed for some other mutants at the predicted interface might result from the mutations not being disruptive enough and thus, do not necessarily disprove the predicted interface.

Despite these limitations, we were able to assess the validity of seven interface predictions using experimentation. We discovered a likely novel DMI type that mediates binding between PEX3 and PEX16, and proposed a model for how PEX3, PEX16, and PEX19 form a trimeric complex at the peroxisomal membrane. We also validated a variation of the LIG_GYF motif class in SNRPB that mediates binding to GIGYF1 thereby potentially connecting mRNA splicing with posttranscriptional control mechanisms. These results confirm in principle that AF is able to predict novel interface types and that it can be used to extend existing interface type definitions. However, our experimental results also highlight clear limitations of AF predictions. Our data suggests that FBXO28 and STX1B as well as STX1B and VAMP2 interact via coiled-coil interfaces but likely at higher stoichiometries and different conformations than predicted. We confirmed the binding pocket in ESRRG but not the predicted interfaces in PSMC5 and we could not substantiate interface predictions for TRIM37 and PNKP. Highly confident interface predictions were obtained for seven additional PPIs that await experimental validation. In summary, we provided experimental evidence and structural information for PPIs whose disruption is likely associated with neurodevelopmental disorders. This information can be explored in future studies aimed at delineating potential molecular mechanisms causing disease. Our study furthermore laid out clear limitations, perspectives, and future needs in AI-based structure prediction to bring us closer to a fully structurally annotated human protein interactome.

# Methods

## Selection of structures for DMI benchmark dataset

To gather a list of ELM classes with structural evidence and annotate their minimal interacting fragments, we downloaded a dataset of solved structures of all ELM classes from ELM DB on 08.10.2021 (ELM class version 1.4) for instances that are annotated as true positives (Kumar et al, 2022). The structures were subject to a series of manual inspections to check their validity for further analysis. First, since AlphaFold can only model the 20 standard amino acids, we excluded any structures with post-translational modifications in the motif. Second, structures that do not resolve all of the residues in a motif as curated by ELM DB were excluded. Third, we restrict our studies to only binary interactions, so DMIs that require more than two proteins to form the binding interface were excluded. Likewise, DMIs with only intramolecular interaction evidence were excluded. We manually annotated the boundaries of the domains by visual inspection of the structures. After this filtering, we identified 136 structures from distinct ELM classes that formed our DMI benchmark dataset (Dataset EV2).

### Sequence identity of the domains in the DMI benchmark dataset

We took all the binding domains in the DMI benchmark dataset and computed their pairwise sequence identity from a global alignment without gap penalties. Matching residues were given a score of 1, otherwise 0. The sum of these scores was divided by the length of the longer sequence to compute the sequence identity.

## Selection of structures for the DDI benchmark dataset

We randomly selected 80 pairs of Pfam domain types that were described in the 3did resource (Mosca et al, 2014) to be in contact with each other in solved structures in the Protein Data Bank (PDB). We manually inspected all PDB entries listed to contain contacts between instances of a given Pfam domain pair until we found one that we considered a genuine domain-domain interaction. These decisions were primarily based on the number of atomic contacts observed and the validity that two folded domains were interacting with each other. Out of the 80 selected Pfam domain pairs, we identified 48 DDI types and 48 corresponding approved DDI structural instances that we selected for the DDI benchmark dataset. The sequences of the minimal interacting domain regions were manually annotated by visual inspection of the structures and used for prediction. A more detailed description of the curation procedure and information on the pairs will be soon published elsewhere (Geist et al, in preparation).

## Generation of random reference sets with minimal interacting regions

### Mutating motif sequences

Key conserved residues of the motifs in the DMI benchmark dataset were identified computationally using the regular expression of the corresponding ELM class in the ELM DB and SLiMSearch (Krystkowiak and Davey, 2017). The defined positions are any positions in the regular expression that are not wildcards. To mutate the key residues to the ones with opposite physico-chemical properties, we substituted one or two key residues with the ones that are of the largest Miyata distance (Miyata et al, 1979) (Dataset EV2).

### Randomizing pairings of known domain-motif interfaces

To simulate non-binding domain-motif pairs, we randomized the pairings of known domain motif interfaces. As some domain types can bind to motifs from distinct ELM classes, we manually checked

that the randomized pairings did not coincide with actual domain-motif interface types (Dataset EV2).

### Randomizing pairings of known domain-domain interfaces

The pairings between known domain-domain interfaces were randomized to form the random reference set for DDIs.

## Generation of positive DMI reference set with fragment extensions

Among the 136 solved structures that we selected previously, we further filtered for structures that consist of only human proteins. To test the potential effect of extension on DMIs that were predicted with different accuracies in their minimal forms, we selected 12 DMI types from the correct sidechain category, 8 DMI types from the correct backbone category and 11 DMI types from the correct pocket category as determined using the motif RMSD calculation. In total, 31 DMI types were selected for extension. Three additional DMI types were originally selected but later on discarded because they contained secondary motif occurrences complicating data analysis. The extensions were done on the canonical sequence of the proteins used to solve the structure. Motif extension 1 extended the motif sequence at both N and C termini by n residues where *n* is the length of the known motif. Motif extension 2 further extended the motif sequence by another *n* residues at both termini. Motif extension 3 and 4 each extended the motif sequence by 2*n* residues at both termini. Motif extension 5 extended the motif sequence by including neighboring domains and motif extension 6 used the full-length protein sequence. On the domain side, domain extension 1 extended the domain sequence to include the disordered regions N- and C-terminally of the binding domain until it reached neighboring domain(s) boundaries. Domain extension 2 included the sequence region of the neighboring domains and domain extension 3 used the full-length protein sequence. In cases where the known motif or binding domain is at the C terminus, we extended the motif or domain sequence on only the N terminus and vice versa. There were some cases where the last extension steps, motif extension 6 and domain extension 3, extended the protein minimally (<20 residues N or C terminal to the previous extension step). These cases were excluded from the analysis. The dataset of extended DMIs is in Dataset EV5. In total, 709 fragment pairs were submitted to AlphaFold. From these, 632 and 616 were successfully modeled by AF v2.2 and v2.3, respectively.

## Generation of random DMI reference set with fragment extensions

To generate a random reference set using the extensions, we randomized the pairings of the 34 DMI types that we selected for extensions and paired their extensions for prediction. Motif extension 6 and domain extension 3 were excluded from the pairing. The dataset of DMIs with random pairings and their extensions can be found in Dataset EV6. In total, 612 predictions were generated, among which 566 and 522 predictions were successfully modeled by AF v2.2 and v2.3, respectively. Since motif extension 6 and domain extension 3 were excluded from the random reference set using the extensions, we also excluded them from the positive reference set extensions during ROC analysis.

This resulted in 563 and 540 predictions from the positive reference set extensions for AF v2.2 and v2.3, respectively.

## Selection of reference datasets for comparison of AF v2.2 with v2.3

All predictions for the minimal DMIs and the random DMIs involving minimal fragments were successfully modeled by both versions of AF. Some extensions from the positive reference set were not successfully modeled by AF v2.2 and v2.3 due to failure from HHblits. To compare AF v2.2 with v2.3, we used only predictions that were successfully modeled by both versions of AF. This resulted in 616 predictions from the extensions of the positive reference set.

## Evaluation of AF sensitivity and specificity when using the fragmentation approach

Among the 34 DMIs selected for extension, we further selected 20 DMIs and retrieved the PPIs mediating these DMIs as the PRS and randomized their pairing to form random domain-motif protein pairs as the RRS. The 20 PPIs from the PRS and the 20 protein pairs from the RRS were subjected to the fragmentation approach, generating 8943 fragment pairs and 11,045 fragment pairs for the PRS and RRS, respectively. All fragment pairs from the PRS and all but one fragment pair from the RRS resulted in an AlphaFold model. Models were deemed highly confident, if the disordered fragment had a motif interface pLDDT of ≥70 or, in case of ordered-ordered models, the average interface pLDDT scored ≥70. To evaluate the sensitivity of the fragmentation approach, we considered all models that met the above mentioned cutoffs and which contained the motif and domain sequence. We superimposed the models onto the corresponding native structures using the minimal domain and computed the RMSD between the minimal motif residues in the native and modeled structure. A model was deemed accurate if the motif RMSD was ≤5 Å. At this cutoff the backbone of the native and modeled motif are well aligned but not necessarily their side chains (see also RMSD subsection below). We repeated the same procedure for each DMI protein pair using full length sequences as input into AF for modeling. In 18 cases AF did not return a model when using full length sequences. Here, we used the largest protein fragments instead for which AF returned a model. Information on the protein pairs, prediction results, and statistics is available in Dataset EV9.

## AlphaFold versions and runs

We used local installations of AlphaFold Multimer version 2.2.0 and 2.3.0 (preprint:Evans et al, 2021) for all protein complex predictions with the following parameters:

 --max_template_date=2020-05-14

 --db_preset=full_dbs

 --use_gpu_relax=False

For every AlphaFold run, five models were predicted with single seed per model by setting the following parameter:

 --num_multimer_predictions_per_model=1

The databases queried during AlphaFold predictions were specified following the instructions from the github page of AlphaFold

([https://github.com/deepmind/alphafold#running-alphafold](https://github.com/deepmind/alphafold#running-alphafold)):

For running AlphaFold Multimer v2.2, the following databases were queried:

--bfd_database_path=bfd_metaclust_clu_complete_id30_c90_-final_seq.sorted_opt

--mgnify_database_path=alphafold_v220_databases/mgy_clusters_2018_12.fa

--obsolete_pdbs_path=alphafold_v220_databases/pdb_mmcif/obsolete.dat

--pdb_seqres_database_path=alphafold_v220_databases/pdb_seqres/pdb_seqres.txt

--template_mmcif_dir=alphafold_v220_databases/pdb_mmcif/mmcif_files

--uniprot_database_path=alphafold_v220_databases/uniprot/uniprot.fasta

--uniclust30_database_path=alphafold_v220_databases/uniclust30/uniclust30_2018_08/uniclust30_2018_08

--uniref90_database_path=alphafold_v220_databases/uniref90/uniref90.fasta

For running AlphaFold Multimer v2.3, the following databases were queried:

--bfd_database_path=alphafold_v230_databases/bfd/bfd_metaclust_clu_complete_id30_c90_final_seq.sorted_opt

--mgnify_database_path=alphafold_v230_databases/mgnify/mgy_clusters_2022_05.fa

--obsolete_pdbs_path=alphafold_v230_databases/pdb_mmcif/obsolete.dat

--pdb_seqres_database_path=alphafold_v230_databases/pdb_seqres/pdb_seqres.txt

--template_mmcif_dir=alphafold_v230_databases/pdb_mmcif/mmcif_files

--uniprot_database_path=alphafold_v230_databases/uniprot/uniprot.fasta

--uniref30_database_path=alphafold_v230_databases/uniref30/UniRef30_2021_03

--uniref90_database_path=alphafold_v230_databases/uniref90/uniref90.fasta

To test the effect of template use on prediction accuracy, the following parameter setting was used to switch off the use of templates during the prediction:

--max_template_date=1950-01-01

For the fragmentation approach, the multiple sequence alignments (MSAs) of a given protein fragment can be reused in subsequent runs where the same fragment is involved. The MSAs were first moved to the prediction output folder and the following parameter was added to enable the reuse of MSAs.

--use_precomputed_msas=True

For efficient computing, we segregated the MSA generation part by using only the CPUs and the model fitting part using the GPUs.

## Calculation of metrics for structural models

### Motif RMSD

We used the software PyMOL (TM) Molecular Graphics System, Version 2.5.0. Copyright (c) Schrodinger, LLC., for the superimposition of AlphaFold models with corresponding solved structures. First, we used the align command to align the domain chain in AlphaFold models with the domain chain in the solved structure. Then, we used the rms_cur command to calculate the all-atom RMSD between the motif chain in AlphaFold models and the motif chain in the solved structure. To ensure that the RMSD calculation was done based on all atom identifiers and without any outlier rejection refinement, the arguments of the rms_cur command, matchmaker and cycles, were set to 0. Prediction accuracy categories were defined based on motif RMSD cutoffs: RMSD ≤ 2 Å for correct sidechain, between 2 Å and 5 Å for correct backbone, between 5 Å and 15 Å for correct pocket and >15 Å for wrong pocket.

### DockQ

The calculation of DockQ scores of AlphaFold models was done in reference to their solved structures using the code available on the github repository of DockQ ([https://github.com/bjornwallner/DockQ](https://github.com/bjornwallner/DockQ), (Basu and Wallner, 2016). DockQ classification was done using the cutoffs provided by DockQ (DockQ: <0.23 for incorrect, between 0.23 and 0.49 for acceptable, between 0.49 and 0.80 for medium and ≥0.80 for high).

### pDockQ

The calculation of pDockQ of AlphaFold models was done by adapting the code available on the github repository from the Elofsson lab ([https://gitlab.com/ElofssonLab/FoldDock/-/blob/main/src/pdockq.py](https://gitlab.com/ElofssonLab/FoldDock/-/blob/main/src/pdockq.py), (Bryant et al, 2022)). The pDockQ score is created by fitting a sigmoidal curve to the DockQ scores of a series of AlphaFold predicted models. The score takes into account the number of interface contacts as well as their pLDDT scores. Of note, the calculation of pDockQ score takes Cβs (Cα for glycine) from different chains within 8 Å from each other as interface contacts which is different from our interface definition (see the subsection below *Domain chain and motif chain interface pLDDT and average interface pLDDT*).

### iPAE

The calculation of iPAE of AlphaFold models was done by adapting code available on the github repository [https://github.com/fteufel/alphafold-peptide-receptors/tree/main](https://github.com/fteufel/alphafold-peptide-receptors/tree/main) (Teufel et al, 2023). The iPAE is the median predicted aligned error at the interface. The authors consider residues in contact if their distance is below 0.35 nm (3.5 Å). The iPAE score could not be calculated for models generated by AlphaFold Multimer version 2.3.0 due to JAX dependency of the pickle files generated by AlphaFold Multimer version 2.3.0.

### Model confidence

The model confidence of AlphaFold models was extracted from the ranking_debug json file. The model confidence is a weighted combination of pTM and ipTM to account for both intra- and interchain confidence:

$$model\ confidence = 0.8 \cdot ipTM + 0.2 \cdot pTM$$

### Domain chain and motif chain interface pLDDT and average interface pLDDT

Since AlphaFold conveniently stores the pLDDT confidence measure for each residue in the B-factor field of the output PDB files, the pLDDT of residues at the interface was parsed from the output PDB files of AlphaFold. Residues at the interface are defined as those that have at least one heavy atom that is less than 5 Å away from any heavy atom of the other chain (calculated using the

PyMOL API). The pLDDT of the residues at the interface from the domain chain and motif chain was averaged to compute the domain chain and motif chain interface pLDDT, respectively. The pLDDT of all the residues from both chains was averaged to compute the average interface pLDDT.

### Residue-residue and atom-atom contacts

Following the interface definition above, the number of unique residue-residue and atom-atom contacts were also quantified as measurements to assess AlphaFold models.

### Mean DockQ between predicted models

The top five models generated by AF, determined based on their model confidence, were considered for computing this metric. To quantify the similarity among the models, we computed DockQ scores between all possible pairs of models by taking the higher ranked model as the "template" model and lower ranked model as the "predicted" model. The mean of these DockQ scores is taken as the similarity among the models in a given prediction. This calculation was done for AF models of minimal DMIs and their randomizations for ROC analysis. The data were stored in Dataset EV2.

## Quantification of motif properties

### Motif hydropathy score and symmetry score

By referring to the Kyte-Doolittle hydrophobicity scale, (Kyte & Doolittle, 1982) the hydropathy scores of the amino acids in a given motif were summed and averaged to compute the average hydropathy of the motif. The average motif symmetry score was computed by taking the sum of the absolute difference of hydropathy scores between motif position n and n - motif length + 1 and division of this sum by half of the motif length:

$$Peptide\ symmetry\ score = \frac{\sum_{n=1}^{a} |(H_n - H_{x-n+1})|}{a}$$

where x is the length of the motif and a is the floor division of x by 2.

### Motif probability

The motif probability reflects the degeneracy of a given motif class as quantified by its regular expression that is annotated in the ELM DB. The motif probability was retrieved from the ELM DB version 1.4.

### Secondary structure elements of motifs

We extracted the secondary structure elements of motifs using the PyMOL API. In cases where the motif adopts partial secondary structure, such as loop-helix-loop or loop-strand-loop, they are treated as helical or strand, respectively.

## Selection of motif classes from ELM DB without annotated structural instances and prediction with AF

By querying the ELM DB for all ELM classes, we retrieved a list of ELM classes and the number of instances with a structure solved (column #instances_in_PDB). We filtered for ELM classes with 0 instances_in_PDB and selected 205 instances out of the filtered ELM classes for

AF prediction. The ELM instances were extended at both N and C termini by n residues where *n* is the length of the ELM instance, according to the benchmarking results. The minimal binding domains of the ELM instances were detected in the interaction partner using Pfam HMMs (Mistry *et al*, 2021). As the domain boundaries detected by Pfam HMMs could be inaccurate, we also extended the domain sequence at the N and C terminus by 20 residues to ensure that the whole folded region was covered. The predictions were performed using AF version 2.3.0. To select a subset of these motif classes, where we can do experimental testing, we also used the InParanoid resource (Persson & Sonnhammer, 2023) to map ELM instances where both proteins are from mouse to their human orthologs. To verify that they indeed do not have structural homologues in the PDB, we both used the SIFTS mapping (Dana *et al*, 2019) between the Pfam domain in ELM and the PDB and also looked at the ELM classes that were listed as homologs on the ELM website.

## Evaluation of effect of fragment extensions on AF prediction accuracies

We superimposed the AF models generated with DMI extensions onto the corresponding solved DMI structures to quantify AF prediction accuracy using motif RMSD calculations. To this end, we aligned the two structures on their minimal binding domains and calculated the all-atom RMSD between the minimal motif in the extension AF model and the minimal motif in the solved structure. To determine potential differences in DMI prediction accuracy when using minimal versus extended protein fragments, we computed the log2 fold change of the all-atom motif RMSD before and after extension.

$$Fold\ change\ in\ prediction\ accuracy = log_2\left(\frac{all\ atom\ RMSD\ motif_{minimal\ DMI}}{all\ atom\ RMSD\ motif_{extended\ DMI}}\right)$$

## Fragment design and fragment pairing for fragmentation approach

We first inspected the monomeric structural models from the AlphaFold database (Varadi et al, 2022; Jumper et al, 2021) of both interacting proteins to determine the boundaries of their ordered and coiled-coil regions, which were also treated as "ordered". All regions that were not annotated as ordered were annotated as disordered. In some cases, an extended loop with low pLDDT can be found within an ordered region. As they can also potentially carry a motif or mediate interactions in another way, these regions were also annotated as disordered in addition to their annotation as being part of a larger ordered region. The disordered regions of the proteins were fragmented into fragment sizes of 10, 20 and 30 residues. To allow AF to sample continuous sequences, we also generated another set of fragments of same sizes that overlap with the previous fragments by sliding the sequence by half the size of the fragment. The unfragmented disordered regions, as well as their fragments, from one protein were then paired with the ordered regions from its interacting partner and vice versa for prediction. The ordered regions from both proteins were also paired for prediction. We decided to manually define boundaries between ordered and disordered regions because testing available code developed for this purpose, like clustering using the PAE matrix,

turned out to be too inaccurate. We observed that erroneous removal of residues close to the domain borders that are still contributing to the folding of a structured domain, can heavily mislead AF predictions.

## Selection of NDD proteins

A list of NDD genes was assembled using whole exome and whole genome sequencing studies of cohorts of NDD patients from Gene4Denovo (Zhao et al, 2020) and Deciphering Developmental Disorders (DDD) study (Firth et al, 2011), respectively. From Gene4Denovo, we selected genes linked to autism-spectrum disorders (ASD), intellectual disability (ID), epilepsy (EE), undiagnosed developmental disorders (UDD) and NDDs in general. Genes with non-coding mutations as well as genes with a false discovery rate (FDR) >= 0.05 were excluded. Similarly, in the DDD study, genes associated with developmental disorders with a neurological component, as well as genes found to be mutated in at least three children with NDDs (labeled as confirmed genes) were retained. The final list included 984 NDD-risk genes. We filtered the HuRI network (Luck et al, 2020) for interactions mediated exclusively by proteins from this NDD gene list resulting in 67 PPIs excluding self-interactions. Since our fragmentation approach generates many fragments, we did not consider PPIs involving proteins that are more than 1500 amino acids in length, resulting in a final list of 62 PPIs that were subjected to AF modeling.

## Manual inspection of interface predictions for NDD-NDD PPIs and selection for experimental validation

Paired fragments from NDD-NDD PPIs were predicted using AF version 2.2 and the prediction results are stored in Dataset EV10. Based on our benchmarking results, we started by manually inspecting all NDD-NDD PPIs that obtained at least one structural model with either a motif chain interface pLDDT of ≥70 for the disordered fragment or with an average interface pLDDT ≥ 70 for structural models with predicted ordered-ordered interfaces (DDIs). However, during the course of these manual inspections, we found that using in addition a model confidence of ≥0.7 for ordered-ordered fragment pairs helped discriminating good from bad structural models. We inspected the ranked_0 models for all fragment pairs that met the above cutoffs but also inspected models scoring somewhat below these cutoffs. For every NDD-NDD PPI we used Interactome3D (Mosca et al, 2013) and PDB database searches (https://www.rcsb.org/ (Berman et al, 2000)) to identify whether a structure already existed for this PPI. In our evaluation of the structural models we also considered if a certain interface was recurrently predicted for different overlapping fragments because this usually hints at increased confidences for the correctness of the interface prediction. We furthermore explored the number and kind of residue-residue contacts predicted by AF by visual inspection of the structural models using PyMol. We searched for functional annotations and existing structures for the monomers using the PDB, ProViz (Jehl et al, 2016), SMART (Letunic et al, 2021), and the scientific literature to identify enzymatic pockets or binding interfaces for DNA, RNA, or metal ions. Observations and justifications for the final evaluation of the predictions for every NDD-NDD PPI are provided in Appendix Supplementary Text S1.

Based on clone availability, we selected 49 of the 62 PPIs for experimental validation of the predicted interfaces using the BRET assay. For 30 of the 49 selected PPIs for experimental testing we obtained sequence-confirmed clones with luciferase and mCitrine fusions. For 28 of these PPIs both partners were expressed in our experimental system as determined by total luminescence and fluorescence measurements (Fig. 3D,F).

## Softwares used

We used the software PyMOL (TM) Molecular Graphics System, Version 2.5.0. Copyright (c) Schrodinger, LLC., for the visualization and superimposition of AlphaFold models.

All codes were written in Python3 and analyses were done using Jupyter notebooks. We used the Python libraries, Biopython (Cock et al, 2009) for sequence similarity computation, pandas (McKinney, 2010) for data analysis, and Matplotlib (Hunter, 2007) and seaborn (Waskom, 2021) for data visualization. ROC and PR statistics were calculated using the Python package sci-kit learn (Pedregosa et al, 2012).

## Cell line culture and maintenance

HEK293 cells were purchased from DSMZ (catalog number ACC305). These cells were grown and maintained in DMEM (Thermo Fisher), supplemented with 10% FBS (PAN-Biotech), 2 mM glutamine (Thermo Fisher) and 1% penicillin–streptomycin (Thermo Fisher). Cells were incubated at 37 °C with 5% $CO_2$. Subcultivation was performed with 1 ml of 0.05% trypsin every 2–3 days for up to 40 passages. For each passage $1–2 \times 10^6$ cells were seeded in T25 flasks (Sarstedt). Then, new cells were thawed from stocks containing $2 \times 10^6$ cells in 1 ml of growth medium, supplemented with 10% DMSO (Sigma). Every 3 months cells were checked for mycoplasma contamination using a PCR test (Dataset EV11). The cell line was purchased from DSMZ four years ago, expanded, aliquoted, and frozen. A new aliquot is thawed after every 40 passages. No further authentication of the cell line has been done.

## Plasmid construction

### Standard controls
The donor and acceptor vectors pcDNA3.1-cmyc-NL-GW (Addgene plasmid ID #113446), pcDNA3.1-GW-NL-cmyc (Addgene plasmid ID #113447), pcDNA3.1 GW-His3C-mCit, pcDNA3.1 mCit-His3C-GW as well as controls pcDNA3.1-NL-cmyc (Addgene plasmid ID #113442), pcDNA3.1-PA-mCit (Addgene plasmid ID #113443) were kindly provided by the Wanker Group (Max-Delbrück-Centrum für Molekulare Medizin, Germany) (Dataset EV12). By default we cloned all ORFs of interest into N-terminal NL and mCit fusion destination vectors and occasionally also transferred ORFs into C-terminal fusion vectors if N-terminal fusions did not result in sufficient BRET signals but the interaction was of high interest to this study and predicted interfaces were closer to the C-terminus. Trepte et al have shown that testing protein pairs in different configurations increases detection rates while maintaining low false detection rates and that BRET signals are higher if fusions are close to the actual interaction interface (Trepte et al, 2018; preprint:Trepte et al, 2021; preprint:Trepte et al, 2023).

## GATEWAY cloning procedure

Full-length wild-type human open reading frames (ORFs) being cloned in GATEWAY entry vectors from the ORFeome collaboration are stored as bacterial glycerol stocks. (ORFeome Collaboration, 2016)

1. The ORFs were inoculated in 96-well plates (Corning), with each well containing 200 uL of LB medium and 100 µg/ml ampicillin. The plate was incubated at 37 °C and left to shake overnight at 190 rpm.

2. In a 96-well PCR plate (Brand) 10 ng of each selected ORF was used per 50 µl PCR reaction (denaturation at 98 °C for 10 s, annealing at 55 °C for 30 s and extension at 72 °C for 3 min, 30 cycles of amplification) using phusion high-fidelity polymerase (NEB) and primers annealing to the backbone of the plasmid (forward: 5′TTGTAAAACGACGGCCAGTC and reverse: 5′GCCAGGAAACAGCTATGACC).

3. The PCR products (6 µl per well) were confirmed through 96-well E-gel with SYBR (Thermo Fisher, Catalog no G720801) using 25 µl of loading buffer (Thermo Fisher) and 20 µl of E-Gel 96 High range DNA marker (Thermo Fisher).

4. In a 96-well PCR plate 1 µl of each amplified PCR product together with 200 ng of above-mentioned destination vectors were directly used per 10 µl LR reaction using 4x LR clonase (Invitrogen), thereby generating expression vectors.

5. The full 10 µl of LR reaction was transformed into chemically competent DH5a cells (30 µl) in a 96-well PCR plate, then recovered in 80 µl of pre-warmed SOC medium at 37 °C for 1 h without shaking.

6. 70 µl of transformed bacteria was plated on 48-well square agar plates and incubated at 37 °C overnight.

7. Afterwards, colonies were selected and inoculated into a 96 deep-well plate containing 2 ml of LB medium and 100 µg/ml ampicillin. The plate was then incubated at 37 ˚C with continuous shaking at 700 rpm in the incumixer for 24 h.

8. The amplified vectors were extracted from the inoculated culture using Plasmid Plus 96-well Miniprep kit (Qiagen). The concentration of each vector was measured with a Nanophotometer and diluted to 100 ng/µl. Next, 600 ng of insert was used for full-length sequencing using the backbone primers (tag-specific NanoLuc forward: 5′GAACGGCAACAAAATTATCGAC, mCitrine forward: 5′AGCAGAATACGCCCATCG and reverse: 5′GGCAACTAGAAGGCACAGTC) and ORF-specific primers (Dataset EV11) to fully cover the ORFs where it was needed (Dataset EV12). All sequence-confirmed ORF sequences used in this study are available in Dataset EV13.

## Site-directed mutagenesis

The primers were manually designed using the following criteria:

1. For point mutation the primers should overlap the site of mutation. The overlap should be 15–20 nucleotides (nt).

2. For the deletion the primers should be designed to exclude the deletion site, but still overlap and the overlap should be as mentioned in step 1.

3. Primer length should be in the range of 32–36 nt.

4. GC content should be between 40–60%.

5. Difference in melting temperature of primers should not exceed 5 °C.

6. The primer ideally should start and end with guanine or cytosine.

7. The designed oligos were grouped by annealing temperature for the next step.

8. In 96-well PCR plate 10 ng of DNA template together with oligos were used per 50 µL of PCR reaction (denaturation at at 98 °C for 2 min, annealing for 15 s and extension at 72 °C for 5 min, 25 cycles of amplification) using phusion high-fidelity polymerase (NEB).

9. 1 µL of DpnI (NEB) was added to the plate with PCR products and incubated at 37 °C for 1 h. The reaction was stopped at 65 °C for 20 min.

10. The PCR products (6 µl per well) were confirmed through 96-well E-gel with SYBR (Thermo Fisher, Catalog no G720801) using 25 µl of loading buffer (Thermo Fisher) and 20 µl of E-Gel 96 High range DNA marker (Thermo Fisher).

11. 3 µL of digested PCR product was transformed into chemically competent DH5a cells (30 µL) in a 96-well PCR plate, then recovered in 80 µL of pre-warmed SOC medium at 37 °C for 1 h without shaking.

12. 70 µL of transformed bacteria was plated on 48-well square agar plates and incubated at 37 °C overnight.

13. Afterwards, colonies were selected and inoculated into a 96 deep-well plate containing 2 ml of LB medium and 100 µg/ml ampicillin. The plate was then incubated at 37 ˚C with continuous shaking at 700 rpm in the incumixer for 24 h.

14. The amplified vectors were extracted from the inoculated culture with Plasmid Plus 96-well Miniprep kit (Qiagen). The concentration was measured with a Nanophotometer and diluted to 100 ng/µl. Next, 600 ng of insert was used for full-length sequencing using primers covering the mutation and ORF-specific primers (Dataset EV11) to fully cover the ORF length (Dataset EV12).

## BRET assay

### Transfection

HEK293 cells were grown and maintained in high-glucose (4.5 g/l) DMEM (Thermo Fisher) for BRET assays. Media was supplemented with 10% fetal bovine serum (PAN-Biotech) and 1% Penicillin/Streptomycin. Cells were grown at 37 °C, 5% $CO_2$, and 85% RH. Cells were subcultured every 2–3 days and transfected with lipofectamine 2000 transfection reagent (Invitrogen) in Opti-MEM medium (Thermo Fisher) using the reverse transfection method according to the manufacturer's instructions. For transfections, cells were seeded at a density of $4.0 \times 10^4$ cells per well in a white 96-well microtiter plate (Greiner) in phenol-red-free, high-glucose DMEM media (Thermo Fisher) supplemented with 5% fetal bovine serum (Thermo Fisher). Transfections were performed with a total DNA amount of 200 ng per well. If the expression plasmid concentration amount was below 200 ng/well, pcDNA3.1 (+) was used as a carrier DNA to reach the total amount of DNA of 200 ng. All protein pairs were tested in both N-terminal fusion orientations (NL-A with mCit-B and NL-B with mCit-A). The following proteins were also tested as C-terminal fusions: CSNK2B-NL, ESRRG-NL, CUL3-NL, PEX3-NL, PEX19-NL, PSMC5-NL, PEX3-mCit, PEX19-mCit, PEX16-mCit, RORB-mCit, ESRRG-mCit, PAX6-mCit, CSNK2B-mCit, PSMC5-mCit, KCTD7-mCit (Dataset EV12).

### Measurement

The plate was incubated 2 days at 37 °C, 5% $CO_2$, and 85% RH before measurements. All measurements were done with the Infinite M200 Pro microplate reader (Tecan). First, 100 µl of the medium was aspirated from each well. The mCitrine fluorescence (FL) was measured in intact cells (excitation/emission 513 nm/548 nm) using a gain of 100. On rare occasions, the plate reader recorded an overflow with these settings (i.e. for GIGYF1 constructs). In these cases, we repeated the measurement with optimal gain settings and used a fluorescein control to normalize fluorescence signals measured with different gain settings. For this purpose, Fluorescein was obtained from Sigma-Aldrich (Catalog No 46955-250MG-F) and used without further purification. A stock solution of Fluorescein (1 mg/ml in Ethanol) was prepared by dissolving 1.3 mg Fluorescein in 1.3 ml absolute ethanol. 100 µl of a 20 µg/ml solution of Fluorescein were added to an empty well immediately before starting the fluorescence measurements. The 20 µg/ml solution of Fluorescein was obtained by preparing a 1:50 dilution in water of the stock solution. After measuring the fluorescence, coelenterazine-h (PJK Biotech GmbH) was added to a final concentration of 5 µM. The cells were briefly shaken for 15 s and incubated for 15 min inside the plate reader at 37 °C. After incubation, total luminescence was measured first followed by short-wavelength (WL) and long-wavelength luminescence (LU) measurements using the BLUE1 (370–480 nm) and the GREEN1 (520–570 nm) filters at 1000 ms integration time. Corrected BRET ratios were calculated as described in (Trepte et al, 2018). Briefly, for every transfected protein pair NL-A and mCit-B, the following two control pairs were measured: NL-Stop with mCit-B and NL-A with mCit-Stop. The maximal BRET from both control pairs was subtracted from the actual test pair to correct for donor bleedthrough, unspecific binding to the tags, and background signal.

### Determination of binding events in BRET assay

To determine whether a protein pair interacted in the BRET assay or not, we used donor:acceptor DNA transfection ratios of 2:50 ng in all cases except for PEX3-PEX16 where we used 8:25 and PEX3:PEX19 where we used 8:50 ng DNA ratios due to low expression levels of PEX3 and a degradation effect of higher PEX16 protein levels on PEX3 expression levels. We requested that cBRETs determined at these transfection ratios were ≥0.05, fluorescence measurements representing mCitrine fusion expression levels to be ≥500 units, and total luminescence measurements representing NL fusion expression levels to be ≥50,000.

### Saturation assay

For donor saturation experiments various donor DNA amounts (1, 2, 4 and 8 ng) encoding NL-fused proteins were co-transfected with increasing amounts of acceptor DNA (12.5, 25, 50, 100, 200 ng) encoding mCitrine-fused proteins. Fluorescence, total luminescence, and BRET measurements were done as described before. BRET measurements were corrected for bleedthrough using NL-Stop transfections. Fluorescence and total luminescence measurements were corrected for background signal using transfections with pcDNA3.1(+) and subsequently used to estimate amounts of expressed proteins and to plot acceptor/donor ratios on the *x*-axis of titration plots.

### Fitting of titration curves

Titration curves were fitted using the leastsq function from the scipy.optimize python package (Virtanen et al, 2020) using the model BRET = ((A/D) * BRETmax)/(BRET50 + (A/D)) described in (Drinovec et al, 2012), which assumes a 1:1 binding mode, to obtain estimates for the BRETmax and BRET50. Standard errors of the BRET50 estimates were obtained from the variance-covariance matrix, calculated by multiplying the fractional covariance matrix (output by leastsq function) by the residual variance. Measuring BRET signals in intact cells for increasing acceptor/donor protein expression ratios results in an eventual saturation of the signal. Fitting this curve allows extraction of the maximal BRET that can be reached and the BRET50, which is the acceptor/donor ratio at which half of the maximal BRET is obtained. The BRET50 is indicative of binding affinity, in analogy to the IC50, however, its accurate estimation requires saturation of the BRET to be observed in the experimental system, which cannot always be achieved because of limited amounts of DNA that cells can be transfected with. Alternatively, if mutations are unlikely to change the overall structure of the fusion constructs and do not alter expression levels compared to wildtype, single point BRET measurements at acceptor/donor ratios prior to BRET saturation are also indicative of changes in binding strength. The BRET titration curves that we obtained for the PNKP-TRIM37 interaction clearly deviated from the assumed 1:1 binding mode because at higher acceptor:donor ratios we observed a sudden increase in BRET again contrary to an expected saturation. The model could thus not be fitted to the titration data.

## Antibodies

Purified anti-HA.11 Epitope Tag, Clone: [16B12], Mouse, Monoclonal (Biolegend, BLD-901502), 1:2000.

Purified anti-GIGYF1, Rabbit, Polyclonal (BETHYL laboratories, Cat. #A304-132A-1), 1:1000.

GAPDH Loading Control Monoclonal Antibody (GA1R), HRP-coupled (Thermo Fisher Cat. MA515738HRP), 1:3000.

## Co-immunoprecipitation and western blot

Snrpb (full-length) and C-terminal truncation mutant (amino acids 1-190) was cloned from mouse cDNA and ligated into pFRT-TO destination plasmid using AscI and PacI restriction sites. The constructs additionally contain C-terminal 2xHA and mNeonGreen tags. Flp-In™ T-REx™ 293 Cell Lines (Thermo Fisher, catalog number: R78007) expressing Snrpb endogenously from a single locus were generated according to the manufacturer's instructions. In brief, pFRT-TO and pOG44 plasmids were co-transfected and hygromycin-resistant colonies were grown, picked and expanded. The Snrpb transgene expression was validated by western blot, RT-qPCR, and immunofluorescence, which showed that ectopic Snrpb-HA was expressed at levels highly similar to the endogenous Snrpb protein.

For the co-immunoprecipitation experiments, $8 \times 10^6$ cells were seeded in a 10 cm dish. The following day, expression of Snrpb-HA was induced by adding 0.1 µg/mL Doxycycline (D9891, Sigma Aldrich) to the culture medium. Parental cells not expressing any HA-tagged transgene were used as a negative control of immunoprecipitation. The next morning the cells were harvested by scraping in culture media, followed by centrifugation and a

single wash in ice-cold PBS. The whole cell extract was prepared by 15 min incubation on ice with 0.3 mL of lysis buffer (200 mM NaCl, 50 mM HEPES, pH 7.6, 0.1% IGEPAL, 10 mM MgCl$_2$, 10% Glycerol, Protease Inhibitor Cocktail (P8340, Sigma Aldrich), Phosphatase Inhibitor (P5726, Sigma Aldrich) followed by 2 cycles of sonication in a Bioruptor Plus (30 s on, 30 s off) and centrifugation for 20 min at $16,000 \times g$. The extract was quantified by a Bradford assay and 1 mg was used for immunoprecipitation, for which the NaCl concentration was adjusted to 100 mM final concentration by diluting with an equal volume of Lysis Buffer containing 0 mM NaCl. 0.05 mg was set aside as input control (5%). 0.02 mL of Thermo Scientific™ Pierce™ Anti-HA Magnetic Beads (Thermo Fisher Cat. 13464229) were incubated with 1 mg protein extract for 1 h at 4 °C on a rotating wheel. The beads were washed three times before eluting the immunoprecipitated proteins with 0.02 mL of 1 x NuPAGE™ LDS Sample Buffer by incubating at 42 °C for 10 min while shaking at 800 rpm. Another 0.01 mL were used for elution, were then combined making a total of 30 μL, which were transferred to a fresh tube and to which 3 μL of 1 M DTT were added. Input and immunoprecipitated eluates were then separated on a 10% Tris-Glycine SDS PAGE using 1xMOPS buffer, immunoblotted on 0.45 μm PVDF membranes (Tris-Glycin Transfer Buffer, 10% Methanol, 300 mA, 1 hour), blocked with 5% milk in TBS-0.2% Tween for 30 min at RT. Primary antibodies were incubated overnight at 4 °C on a rocker followed by washes and incubation with secondary HRP-labeled antibodies (1 h at RT in 5% milk, TBS-0.2% Tween). Blots were developed using Pierce™ ECL Western Blotting Substrate (Thermo Fisher Cat. 32209) or SuperSignal West Femto Maximum Sensitivity Substrate Kit (Thermo Fisher Cat. 34095) and imaged on a ChemiDoc MP V3 (Bio-Rad). The cell line was authenticated via X-Gal staining, qPCR and Sanger Sequencing.

## Data availability

The datasets and computer code produced in this study are available in the following databases:

- Interaction data: submitted to the IMEx (http://www.imexconsortium.org) consortium through IntAct (Del Toro et al, 2022) and assigned the identifier IM-29904.
- Computer scripts for data processing and analysis: available at GitHub under https://github.com/KatjaLuckLab/AlphaFold_manuscript.

## Peer review information

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

## Acknowledgements

We thank all members of the Luck, Gibson, and Schueler-Furman labs as well as Julian König and Anton Khmelinskii for helpful discussions and input. We thank Izabella Krystkowiak and Norman Davey for helping us access the SLiMSearch resource with an API. We thank Fridolin Kielisch for advice on statistical analysis as well as the media lab and protein production core facilities of IMB. Support from IMB's IT department and especially help from Christian Dietrich for local installations of AlphaFold is gratefully acknowledged. The GPU cluster on which part of the AlphaFold predictions were performed was funded by the Ministry of Science and Health (MWG), Rhineland Palatinate (funding ID: TB-Nr.:3658/19). We are very thankful for support from EMBL IT Services and the HPC resources for running AlphaFold predictions for this project. This work is funded by the Deutsche Forschungsgemeinschaft (DFG, German Research Foundation) – Project-IDs LU 2568/1-1 and SFB1551 Project No 464588647 awarded to KL. JS acknowledges a PhD stipend from IMB's collaborative research initiative. JKV was supported by the European Union's Horizon 2020 UBIMOTIF programme (860517). This work was supported, in whole or in part, by the Israel Science Foundation, founded by the Israel Academy of Science and Humanities (grant number 301/2021 to OS-F).

## Author contributions

**Chop Yan Lee**: Data curation; Formal analysis; Investigation; Visualization; Methodology; Writing—original draft; Project administration; Writing—review and editing. **Dalmira Hubrich**: Data curation; Formal analysis; Investigation; Visualization; Methodology; Writing—original draft; Writing—review and editing. **Julia K Varga**: Data curation; Formal analysis; Investigation; Visualization; Writing—original draft; Writing—review and editing. **Christian Schäfer**: Data curation; Investigation; Methodology. **Mareen Welzel**: Investigation. **Eric Schumbera**: Methodology. **Milena Djokic**: Data curation. **Joelle M Strom**: Formal analysis; Investigation; Visualization. **Jonas Schönfeld**: Investigation. **Johanna L Geist**: Investigation. **Feyza Polat**: Investigation. **Toby J Gibson**: Resources; Supervision; Writing—review and editing. **Claudia Isabelle Keller Valsecchi**: Supervision; Funding acquisition; Investigation; Writing—review and editing. **Manjeet Kumar**: Resources; Formal analysis; Methodology; Writing—review and editing. **Ora Schueler-Furman**: Conceptualization; Supervision; Funding acquisition; Writing—original draft; Writing—review and editing. **Katja Luck**: Conceptualization; Data curation; Formal analysis; Supervision; Funding acquisition; Investigation; Visualization; Methodology; Writing—original draft; Project administration; Writing—review and editing.

## Disclosure and competing interest statement

The authors declare no competing interests.

