## [Peer Review File · Molecular Systems Biology]

Systematic discovery of protein interaction interfaces using AlphaFold and experimental validation

Chop Yan Lee, Dalmira Hubrich, Julia Varga, Christian Schäfer, Mareen Welzel, Eric Schumbera, Milena Đokić, Joelle Strom, Jonas Schönfeld, Johanna Geist, Feyza Polat, Toby Gibson, Claudia Keller Valsecchi, Manjeet Kumar, Ora Schueler-Furman, and Katja Luck

DOI: [10.15252/msb.202311922](https://doi.org/10.15252/msb.202311922)

Corresponding author(s): Ora Schueler-Furman (ora.furman-schueler@mail.huji.ac.il), Katja Luck (k.luck@imb-mainz.de), Ora Schueler-Furman (ora.furman-schueler@mail.huji.ac.il)

Review Timeline:

Submission Date:	3rd Aug 23
Editorial Decision:	30th Aug 23
Revision Received:	27th Oct 23
Editorial Decision:	30th Nov 23
Revision Received:	4th Dec 23
Accepted:	5th Dec 23

Editor: Poonam Bheda

Transaction Report:

30th Aug 2023

Dear Dr. Luck,

Thank you for the submission of your manuscript to Molecular Systems Biology. We have now received feedback from the two reviewers who agreed to evaluate your manuscript. As you will see from the reports below, the referees acknowledge the interest of the study and are overall supporting publication of your work pending appropriate revisions.

Addressing the reviewers' concerns in full will be necessary for further considering the manuscript in our journal, and acceptance of the manuscript will entail a second round of review. Molecular Systems Biology encourages a single round of revision only and therefore, acceptance or rejection of the manuscript will depend on the completeness of your responses included in the next, final version of the manuscript. For this reason, and to save you from any frustrations in the end, I would strongly advise against returning an incomplete revision.

We are expecting your revised manuscript within three months, if you anticipate any delay, please contact us.

We require:

4) A .docx formatted letter INCLUDING the reviewers' reports and your detailed point-by-point responses to their comments. As part of the EMBO Press transparent editorial process, the point-by-point response is part of the Review Process File (RPF), which will be published alongside your paper.

5) A complete author checklist, which you can download from our author guidelines (<https://www.embopress.org/page/journal/17574684/authorguide#submissionofrevisions>). Please insert information in the checklist that is also reflected in the manuscript. The completed author checklist will also be part of the RPF.

6) Please note that all corresponding authors are required to supply an ORCID ID for their name upon submission of a revised manuscript.

7) It is mandatory to include a 'Data Availability' section after the Materials and Methods. Before submitting your revision, primary datasets produced in this study need to be deposited in an appropriate public database, and the accession numbers and database listed under 'Data Availability'. Please remember to provide a reviewer password if the datasets are not yet public (see <https://www.embopress.org/page/journal/17574684/authorguide#dataavailability>).

In case you have no data that requires deposition in a public database, please state so in this section. Note that the Data Availability Section is restricted to new primary data that are part of this study. This study includes no data deposited in external repositories.

8) For data quantification: please specify the name of the statistical test used to generate error bars and P values, the number (n) of independent experiments (specify technical or biological replicates) underlying each data point and the test used to calculate p-values in each figure legend. The figure legends should contain a basic description of n, P and the test applied. Graphs must include a description of the bars and the error bars (s.d., s.e.m.). Please provide exact p values.

9) Our journal encourages inclusion of *data citations in the reference list* to directly cite datasets that were re-used and obtained from public databases. Data citations in the article text are distinct from normal bibliographical citations and should directly link to the database records from which the data can be accessed. In the main text, data citations are formatted as

follows: "Data ref: Smith et al, 2001" or "Data ref: NCBI Sequence Read Archive PRJNA342805, 2017". In the Reference list, data citations must be labeled with "[DATASET]". A data reference must provide the database name, accession number/identifiers and a resolvable link to the landing page from which the data can be accessed at the end of the reference. Further instructions are available at .

<https://www.embopress.org/page/journal/17574684/authorguide#expandedview>

11) For more information: There is space at the end of each article to list relevant web links for further consultation by our readers. Could you identify some relevant ones and provide such information as well? Some examples are patient associations, relevant databases, OMIM/proteins/genes links, author's websites, etc...

12) Author contributions: CRediT has replaced the traditional author contributions section because it offers a systematic machine readable author contributions format that allows for more effective research assessment. Please remove the Authors Contributions from the manuscript and use the free text boxes beneath each contributing author's name in our system to add specific details on the author's contribution. More information is available in our guide to authors.

13) Disclosure statement and competing interests: We updated our journal's competing interests policy in January 2022 and request authors to consider both actual and perceived competing interests. Please review the policy <https://www.embopress.org/competing-interests> and update your competing interests if necessary.

14) Every published paper now includes a 'Synopsis' to further enhance discoverability. Synopses are displayed on the journal webpage and are freely accessible to all readers. They include a short stand first (maximum of 300 characters, including space) as well as 2-5 one-sentences bullet points that summarizes the paper. Please write the bullet points to summarize the key NEW findings. They should be designed to be complementary to the abstract - i.e. not repeat the same text. We encourage inclusion of key acronyms and quantitative information (maximum of 30 words / bullet point). Please use the passive voice. Please attach these in a separate file or send them by email, we will incorporate them accordingly.

Please also suggest a striking image or visual abstract to illustrate your article as a PNG file 550 px wide x 300-600 px high. Share synopsis text and image, as well as eTOC:

Please note that these would be the final versions and changes during proofing are usually not allowed

15) As part of the EMBO Publications transparent editorial process initiative (see our Editorial at <http://embomolmed.embopress.org/content/2/9/329>), Molecular Systems Biology Medicine will publish online a Review Process File (RPF) to accompany accepted manuscripts.

In the event of acceptance, this file will be published in conjunction with your paper and will include the anonymous referee reports, your point-by-point response and all pertinent correspondence relating to the manuscript. Let us know whether you agree with the publication of the RPF and as here, if you want to remove or not any figures from it prior to publication.

Molecular Systems Biology has a "scooping protection" policy, whereby similar findings that are published by others during review or revision are not a criterion for rejection. Should you decide to submit a revised version, I do ask that you get in touch after three months if you have not completed it, to update us on the status.

I look forward to receiving your revised manuscript.

Yours sincerely,

Poonam Bheda

Poonam Bheda, PhD
Scientific Editor

Reviewer #1:

In this paper, the authors examine the use of AlphaFold (v2.2 and v2.3) to identify short linear motifs binding to proteins. They conclude that "The structure predictions were highly sensitive but not very specific when using small protein fragments" and that "Sensitivity decreased substantially when using long protein fragments or full length proteins with intrinsically disordered regions". In general, this is an interesting study that highlights important aspects and limitations of AlphaFold. I focus on the analytical method of the paper and not on the biological results (i.e. the predicted binders) as this is outside my areas of expertise.

Major:

1. I would guess (from my own experience) that reliable predictions are possible to detect using a "consensus" methodology. In my experience (not carefully benchmarked, though), correct predictions often result in all models from AlphaFold being very similar, while this is not the case for unreliable predictions. This is not at all explored but should be easy to do (just take the average DockQ for all 5 models compared with each other as we did in PMID:34788800).

It should also be highlighted that many earlier studies (in particular Bryant 2022) used AlphaFold monomer (v2.0) and not AlphaFold-multimer. Therefore, a direct comparison is not possible. I would assume that AlphaFold-multimer has a tendency to put things together even if they do not interact more frequently than AlphaFold2.0 (but this needs to be tested). Also AlphaFold-multimer v2.1 behaved very differently and actually often gave the best results for hard docking cases involving disordered regions, see PMID: 37548092. Although we do not find it necessary to redo the evaluations using all versions of AlphaFold, it should at least be discussed.

Reviewer #2:

In "Systematic discovery of protein interaction interfaces using AlphaFold and experimental validation", Luck and colleagues present an AlphaFold-based approach for creating structural models of domain-motif and domain-domain interactions, including experimental validation of selected cases.

Overall the manuscript is interesting and the approach seems promising for future applications. A great number of details are presented, and while interesting - especially for those studying these proteins, or these classes of interaction, specifically - they do not necessarily make it easier to read the manuscript. We reviewed this in a team effort between a PhD student and a mid-career researcher, and the more junior member felt that some context was missing. What is the state of the art, where exactly does the need for new modelling arise, is this a benchmark-type paper or not? How is the approach different from methods looking for hot spots, fingerprints etc. (MASIF, SBiR). Hence we encourage the authors to consider whom they see as their target audience, and add introductory paragraphs to make sure those from further outside the DMI field can see the big picture and follow along. Possibly also moving some detail into supplementary.

As a specific suggestion, a figure as in S1A would be very helpful, we suggest to move this to the main manuscript.

Minor comments:

P2: "AF2 benchmarking studies reported similarly high success rates for DMI compared to general docking benchmark datasets" - reference?

Fig. 1B - "no significant differences" is technically correct, but the spread certainly differs between classes / sec.struct
What do figures 1.B-1.C contribute to the story? Could they be moved to supplementary?

Fig. 1.E, add a line at 0.5 to remind readers of the random expected performance

Define iPAE and pDockQ metrics somewhere in the manuscript

Regarding the "One mutation" approach: to be the devil's advocate, do you know that these would not bind? To phrase it differently, is the test "would this peptide bind in this position, assuming it does bind" vs. "will this bind in a cell"?

Please provide details on the sequence diversity (pairwise RMSDs) of the proteins chosen for validation, as well as the alignment depths, as this has been shown to be a key influence on AF performance

Fig. 3A, more detail needed in the legend

A and B are the proteins

Could also just refer to the relevant methods section

For motifs where you found that xtal structures existed but were not annotated in ELM, have these now been added to ELM?

(We assume so as the ELM authors are co-authors, but it's not stated)

For other scientists to get the most practical use out of this, actual thresholds would be very helpful

For the pathogenic mutations, please elaborate on your interpretation of the findings, especially those where mutations did not affect binding. Do these mutations affect other functions of the protein? Stability? PTM? Or do you suspect these are not, after all, pathogenic?

P7, neo-N-terminal motif: I suggest rephrasing to highlight the importance of integrating other biological data

The helix extension seems to be yet another instance of AF making helices too long

P8, first lines - do these false binding modes look suspicious upon manual inspection?

P8-9, difficult to follow if the reader is not familiar with BRET, maybe you could provide some more background in supplemental information?

We thank both reviewers for their time and critical feedback, which have clearly helped improve the manuscript.

Reviewer #1:

In this paper, the authors examine the use of AlphaFold (v2.2 and v2.3) to identify short linear motifs binding to proteins. They conclude that "The structure predictions were highly sensitive but not very specific when using small protein fragments" and that "Sensitivity decreased substantially when using long protein fragments or full length proteins with intrinsically disordered regions". In general, this is an interesting study that highlights important aspects and limitations of AlphaFold. I focus on the analytical method of the paper and not on the biological results (i.e. the predicted binders) as this is outside my areas of expertise.

Major:

1. I would guess (from my own experience) that reliable predictions are possible to detect using a "consensus" methodology. In my experience (not carefully benchmarked, though), correct predictions often result in all models from AlphaFold being very similar, while this is not the case for unreliable predictions. This is not at all explored but should be easy to do (just take the average DockQ for all 5 models compared with each other as we did in PMID:34788800).

This is an interesting idea. We computed the proposed metric for our DMI benchmark dataset consisting of the minimal interacting regions for structurally resolved DMIs and random pairings of the domains and motifs. We found that this feature has moderate predictive power. We added this result to page 4 lines 11-14 and Appendix Figure S2C and added a description to the Methods section on page 20 lines 17-24.

It should also be highlighted that many earlier studies (in particular Bryant 2022) used AlphaFold monomer (v2.0) and not AlphaFold-multimer. Therefore, a direct comparison is not possible. I would assume that AlphaFold-multimer has a tendency to put things together even if they do not interact more frequently than AlphaFold2.0 (but this needs to be tested). Also AlphaFold-multimer v2.1 behaved very differently and actually often gave the best results for hard docking cases involving disordered regions, see PMID: 37548092. Although we do not find it necessary to redo the evaluations using all versions of AlphaFold, it should at least be discussed.

We absolutely agree with the reviewer that comparisons of AF performance across studies is problematic because different AF versions and metrics were used. We have already been highlighting this in the introduction in the previously submitted manuscript (page 1, lines 42-47) and have partially addressed this limitation with our study where we have assessed 9 metrics for their ability to discriminate good from bad structural models using two different AF versions and two benchmark datasets representing two different classes of protein interaction interfaces. To further stress that cross-study comparisons of AF performance are problematic we have added text to the discussion section where we relate our findings to previously published ones (page 12, lines 19-22).

Reviewer #2:

In "Systematic discovery of protein interaction interfaces using AlphaFold and experimental validation", Luck and colleagues present an AlphaFold-based approach for creating structural models of domain-motif and domain-domain interactions, including experimental validation of selected cases.

Overall the manuscript is interesting and the approach seems promising for future applications. A great number of details are presented, and while interesting - especially for those studying these proteins, or these classes of interaction, specifically - they do not necessarily make it easier to read the manuscript. We reviewed this in a team effort between a PhD student and a mid-career researcher, and the more junior member felt that some context was missing. What is the state of the art, where exactly does the need for new modelling arise, is this a benchmark-type paper or not?

We rewrote the abstract to improve clarity and add context information. We furthermore removed some details from the introduction (page 1, lines 3-5, page 2 lines 25-31) to focus on the state-of-the-art, open questions, and necessary context of this study:

- the need to more efficiently structurally annotate the human protein interactome (page 1, lines 9-10),

- background information and functional importance of domain-motif interactions and difficulties in studying them (page 1, lines 10-26),
- knowledge gaps in the performance of AF to predict protein interaction interfaces and especially domain-motif interfaces (page 1, lines 31-47),
- Contradictory performance estimates of AF when using full length protein sequences versus short protein fragments as input and the influence of disordered regions of proteins on AF outcome (page 2, lines 2-21),
- Need for experimental methods to validate predicted protein interaction interfaces at scale (page 2, lines 23-31)

On page 2, lines 31-41 we summarize the key findings of this study, stating that we performed benchmarking of AF and found that AF performance drops when using longer protein fragments, developed a prediction pipeline particularly suited for DMI predictions with AF, and the application of this pipeline followed by experimental validation. We hope that this content meets the reviewer's expectation in terms of putting this study in context.

How is the approach different from methods looking for hot spots, fingerprints etc. (MASIF, SBiR).

AlphaFold-Multimer can predict a structure of two or more interacting proteins. It thus goes beyond methods that can detect hot spots for binding or pockets on folded domains. More recently developed, highly sophisticated methods like MASIF can also predict structures of interacting protein fragments but likely much more heavily rely on structures of the monomers being available in the bound conformation. We have added text to the discussion section about this point (page 13, lines 20-24).

Hence we encourage the authors to consider whom they see as their target audience, and add introductory paragraphs to make sure those from further outside the DMI field can see the big picture and follow along. Possibly also moving some detail into supplementary. As a specific suggestion, a figure as in S1A would be very helpful, we suggest to move this to the main manuscript.

In addition to the previously described changes to the abstract and introduction, we also revised the results and discussion section to better place our results in context, focus on key aspects and remove details. We have moved Appendix Figure S1A back as a main figure, we have added some illustrations to Figure 1B and have also expanded the schematic of Figure 2A to better visually explain the assessment of fragment extension on AF performance. We have moved negative results shown in Figure 1B-D to the appendix. We have merged the first and second results section and rewritten parts to improve clarity (page 3, line 39 - page 4, line 11). We have improved transitions, motivations, and conclusions in the 2nd results section (page 4, line 27, page 4, line 39 - page 5, line 26). We removed a rather complicated example from the 4th results section (page 6, line 33) to streamline. We moved the evaluation of AF on domain-domain interfaces here as the 5th section to improve the flow of the manuscript and added text to introduce into this part (page 7, line 2-5). We did major changes to the 7th results section where we apply our AF prediction pipeline to PPIs from the HuRI dataset. We moved experimental details to the methods section (page 7, line 31, page 8 line 12, line 24-25, page 21, line 40 – page 22 line 2, lines 14-17, page 22, lines 41- page 23, line 2). We added information on how we identified likely wrong prediction models from AF (page 8, lines 17-22). We very much shortened the results section on experimental validations of interface predictions for the PPIs PNKP-TRIM37, PSMC5-ESRRG, STX1B-VAMP2, and STX1B-FBXO28 and added them to the 7th results section (page 8 line 33 - page 9, line 33). We kept results for the PPIs PEX3-PEX19, PEX3-PEX16, and SNRPB-GIGY1 as separate sections given the interesting findings we made but shortened and restructured these sections to improve clarity (page 9, line 37 - page 11, line 33).

We rewrote the first paragraph of the discussion to better put our study in the broader context of the field (page 11, line 35 - page 12, line 1). We removed details from the discussion (page 12, line 32, page 13, line 31-46).

Minor comments:

P2: "AF2 benchmarking studies reported similarly high success rates for DMI compared to general docking benchmark datasets" - reference?

We added two references to this sentence (page 2, line 14).

Fig. 1B - "no significant differences" is technically correct, but the spread certainly differs between classes / sec.struct

We added text to describe this (page 3, line 24-25).

What do figures 1.B-1.C contribute to the story? Could they be moved to supplementary?

We moved Figure 1B-D to Appendix Figure S1 and moved previous Appendix Figure S1A back to Figure 1.

Fig. 1.E, add a line at 0.5 to remind readers of the random expected performance

We added this line to Figure 1B and Figure 2H.

Define iPAE and pDockQ metrics somewhere in the manuscript

We already had a description of all computed metrics in the methods section. Since we feel that a description of all the metrics in the first results section would break the flow, we added text in the results section that specifically points the reader to the methods section to obtain information on how the metrics were calculated (page 4, line 3).

Regarding the "One mutation" approach: to be the devil's advocate, do you know that these would not bind? To phrase it differently, is the test "would this peptide bind in this position, assuming it does bind" vs. "will this bind in a cell"?

The reviewer is right that we do not have definitive proof that the introduced mutations to benchmark AF's specificity would be disruptive to binding. However, based on various experimental studies of linear motifs reviewed in Diella et al FrontBiosciences 2008, there is a general sense that mutation of single key-conserved residues in motif instances is disruptive to binding, at least when testing the motif sequence and corresponding domain sequence in in vitro binding assays. We have added text to the results section to clarify our rationale (page 3, line 44-46).

Please provide details on the sequence diversity (pairwise RMSDs) of the proteins chosen for validation,

We performed pairwise sequence alignments of all domain sequences used in the DMI benchmark dataset and found that they mostly share a sequence similarity of 20-30%. We added a histogram of the shared sequence similarity to the manuscript as Appendix Figure S1A and added text to the results section (page 3, lines 8-9) and methods section page 15 lines 28-32.

as well as the alignment depths, as this has been shown to be a key influence on AF performance

Multiple previously published studies have investigated the influence of the alignment depth or level of conservation on AF protein complex prediction outcomes (Bret et al 2023 BioRxiv, Bryant et al Nature Comm 2022, Yin et al Prot Science 2022 to name a few). While some studies observe a certain influence of the alignment depth on model accuracy (Bryant et al Nature Comm 2022) others did not observe a correlation (Yin et al Prot Science 2022). Also the pairing of alignments between both interaction partners was shown to have a limited influence on prediction outcomes (Bret et al 2023 BioRxiv). We focussed in our study on the development of an AF pipeline that can increase DMI prediction successes. The alignment depth is a variable that cannot be influenced when using AF for protein complex prediction and it was unlikely to be an important feature that could help discriminating good from bad structural models. We therefore decided not to explore the role of the alignment depth for AF predictions in this study. Given that the manuscript is already very information-heavy as pointed out by the reviewer, we would kindly argue for these reasons not to add this aspect to the manuscript. We have added text to the discussion where we briefly discuss the potential use of the alignment depth in future studies to identify PPIs for which structural modeling will likely be more successful (page 12, lines 11-14).

Fig. 3A, more detail needed in the legend

A and B are the proteins

Could also just refer to the relevant methods section

We have now described the schematic in Figure 3A more clearly.

For motifs where you found that xtal structures existed but were not annotated in ELM, have these now been added to ELM? (We assume so as the ELM authors are co-authors, but it's not stated)

The missing structures from ELM are now listed in Table EV3 and will be converted to ELM instances and added to the resource as soon as we can, which should be before publication.

For other scientists to get the most practical use out of this, actual thresholds would be very helpful.

We have added optimal cutoffs to Table EV2 where we already reported various ROC statistics and are now explicitly pointing to it in the results section (page 4, line 11). We added information on our most preferred metric and cutoff at the end of the 3rd results section (page 6, lines 5-7) and provide all cutoffs from the DMI extension analysis in Table EV5. We added cutoff information on the benchmarking of the DDI dataset at the end of the 5th results section (page 7, line 11-12) and provide all optimal cutoffs from the DDI benchmarking in Table EV6. We also added optimal cutoff information for the benchmarking of AF v2.3, which can be found in Table EV2, EV5 and EV6. We added text to the results section about the evaluation of de novo interface predictions for HuRI PPIs (page 8, lines 17-22). We added text to the methods section to detail the cutoffs used when evaluating models produced from fragmenting random protein pairs (page 17, lines 27-29). We added text to the methods section describing the cutoffs used when evaluating predictions on HuRI PPIs (page 22, lines 23-28) and when evaluating AF models based on the motif RMSD (page 19, lines 13-15).

For the pathogenic mutations, please elaborate on your interpretation of the findings, especially those where mutations did not affect binding. Do these mutations affect other functions of the protein? Stability? PTM? Or do you suspect these are not, after all, pathogenic?

The fusions needed for the BRET assay fortunately allow at the same time to monitor expression levels of the interaction partners via measurement of total luminescence in intact cells (indicating expression levels of NanoLuc luciferase (NL) fusion constructs) and via measurement of fluorescence in intact cells (indicating expression levels of mCitrine (mCit) fusion constructs). We provide these expression level measurements for all our constructs in the Appendix as supplementary figures. All of the pathogenic mutations tested in this study either did not or only very slightly alter expression levels compared to the wildtype construct (i.e Appendix Figure S2I and S8D). In cases where pathogenic mutations did not result in a change of binding as monitored by the BRET assay, we can of course not exclude that other functionalities of the mutated protein might be altered such as perturbation of other protein interaction interfaces or PTMs. To better guide the reader in the interpretation of our results when testing pathogenic mutations, we added text to the S225N mutation in HCFC1 in the results section (page 5, lines 19-23).

P7, neo-N-terminal motif: I suggest rephrasing to highlight the importance of integrating other biological data

The helix extension seems to be yet another instance of AF making helices too long

Inspection of solved structures showing binding of CASP peptides to the BIR domain and inspection of the structural models generated by AF shows that steric clashes likely prohibit proper docking of the motif into the BIR binding pocket if the motif is extended at its N-terminus. In an attempt by AF to still dock the peptide, it shifts partially out of the pocket and docks it in reversed order. AF did not predict the extended peptide to bind in proper helical conformation. The modeled extended peptide shown now in Figure 2G makes some non-helical kink. We have accordingly extended our description of this case in the results section (page 6, line 37-43) and have moved the figure panel from supplement back as Figure 2G.

P8, first lines - do these false binding modes look suspicious upon manual inspection?

Manual inspection of structural models generated by AF to evaluate their correctness is very time-consuming and can easily take a few hours per PPI because we also consider published literature about the interaction partners and their homologs, especially for structural information and functional sites. We have done this meticulous work for the structural models generated for the 62 HuRI PPIs and report now in the results section (page 8, lines 17-22) and in more detail in Appendix Text S1 (already present in the previous submission) on likely wrong predictions returned by AF. For example, we identify docking of partners into enzymatic sites, sites known to bind nucleic acids or metal ions, and docking of sequences that are remotely similar to sequences known to bind to these pockets. We expect similar observations for the high scoring structural models obtained from submitting fragments of random protein pairs. Because it is unlikely that we would gain additional information from analyzing these likely false models and because of the considerable amount of time it would take, we kindly refrained from doing this additional analysis.

However, revisiting the benchmarking of the fragmentation approach in the light of the reviewer's question, we realized that while we assessed increases in false positive rate when submitting many fragment pairs to AF

for interface prediction, we did not assess increases in sensitivity. To address this, we performed the fragmentation approach on the 20 DMI pairs that were used before to make random protein pairs and assessed how fragmentation would increase AF sensitivity compared to using full length sequences. For 18 protein pairs AF was not able to produce a model when using full length sequences. In these cases we used fragmentation step 5 for motifs (adding neighboring domains) and/or fragmentation step 2 for domains (adding neighboring domains) for interface prediction (Figure 2A). We analyzed all fragment and full length protein pairs that resulted in a model that met the motif chain interface pLDDT cutoff of ≥ 70 . These models were superimposed onto the native structure using the minimal interacting domain sequence and the RMSD was computed on the minimal motif sequence. We plotted the RMSD for all these models and each DMI pair and found that while only for 6 out of the 20 DMI pairs the nearly full length sequences resulted in a model that accurately docks the motif, this was the case for 12 out of 20 DMI pairs when using smaller protein fragments doubling the number of DMI pairs for which accurate models could be obtained. Of note, we considered a model as accurate if the backbone of the motif sequence from the model was well aligned with the backbone of the motif from the native structure. Based on our analysis shown in Figure 1A, this is the case at a RMSD cutoff of 5Å that we used in this analysis. We have modified the results text accordingly (page 7, line 34 - page 8, line 8), added this new analysis as Figure 3B to the manuscript, and put the previous panel Figure 3B showing the results on the random protein pairs to the Appendix as Figure S6D. We also added text to the methods section describing this analysis (page 17 line 29-39). We think that this additional analysis nicely completes this part of the manuscript and improves the story line. We hope that the reviewers feel the same.

P8-9, difficult to follow if the reader is not familiar with BRET, maybe you could provide some more background in supplemental information?

We removed these technical parts from the main text on the PNKP-TRIM37 interaction (page 8 lines 33-43) and added it to the methods section (page 26, line 36 and page 27, lines 1-4). We also point to the methods section from the corresponding figure legend (Figure 4).

30th Nov 2023

Manuscript Number: MSB-2023-11922R

Title: Systematic discovery of protein interaction interfaces using AlphaFold and experimental validation

Dear Dr. Luck,

Thank you again for submitting your work to Molecular Systems Biology. We have now heard back from the two referees who evaluated your study, and I am pleased to inform you that we will be able to accept your manuscript pending the following final amendments:

1) In the main manuscript file, please do the following:

- Please include keywords (up to max. 5).
- Data availability: The Data Availability section needs to be formatted according to the example below:

The datasets and computer code produced in this study are available in the following databases:

- Chip-Seq data: Gene Expression Omnibus GSE46748 (<https://www.ncbi.nlm.nih.gov/geo/query/acc.cgi?acc=GSE46748>)
- Modeling computer scripts: GitHub (<https://github.com/SysBioChalmers/GECKO/releases/tag/v1.0>)
- [data type]: [full name of the resource] [accession number/identifier] ([doi or URL or identifiers.org/DATABASE:ACCESSION])

Please check "Author Guidelines" for more information.

<https://www.embopress.org/page/journal/17574684/authorguide#availabilityofpublishedmaterial>

- Please rename "Conflict of Interest" to "Disclosure and competing interests statement". We updated our journal's competing interests policy in January 2022 and request authors to consider both actual and perceived competing interests. Please review the policy <https://www.embopress.org/competing-interests> and update your competing interests if necessary.

- Author contributions: Please remove it from the manuscript and specify author contributions in our submission system. CRediT has replaced the traditional author contributions section because it offers a systematic machine-readable author contributions format that allows for more effective research assessment. You are encouraged to use the free text boxes beneath each contributing author's name to add specific details on the author's contribution. More information is available in our guide to authors:

<https://www.embopress.org/page/journal/17574684/authorguide#authorshipguidelines>

- Data not shown: We do not allow statements/conclusions with "data not shown". As per our guidelines, on "Unpublished Data" the journal does not permit citation of "Data not shown". All data referred to in the paper should be displayed in the main or Expanded View figures. Please remove from pages 28 and 35.

2) In the Materials and Methods, please take care of the following:

- Cell lines: Please also be sure to include a sentence in the Materials and Methods as to whether or not the cell lines were recently authenticated.

3) Please place individual sections of the manuscript in the following order: Title page - Abstract & Keywords - Introduction - Results - Discussion - Materials & Methods - Data Availability - Acknowledgements - Disclosure and Competing Interests Statement - References - Figure Legends - Expanded View Figure Legends.

4) Appendix file and Expanded View content:

- In the Appendix file, please ensure the word "Appendix" is included in all labels for Appendix Figures and Appendix Tables including in the Table of Contents.

- The Dataset EV1 legend should be removed from Appendix PDF, and included in the zip folder Dataset EV1;

- Tables EV1-EV12 should be renamed to Dataset EV2-EV13 with the corresponding callouts and legends labeled as Dataset EV2-EV13 in each Excel file. These legends should also be removed from the Appendix PDF

5) Data availability: The IM-29904 dataset does not seem to be publicly available. Please be aware that all deposited datasets should be freely accessible prior to publication.

6) Please ensure that all funding sources are entered into the eJP manuscript submission system (i.e. please add the Ministry of Science and Health (MWG), Rhineland Palatinate (funding ID: TB-Nr.:3658/19); PhD stipend from IMB's collaborative research initiative)

7) Synopsis: Please check your synopsis text and image before submission with your revised manuscript. Please be aware that in the proof stage minor corrections only are allowed (e.g., typos).

8) Source Data: The panel labels for Source Data files seem to be off. There are updated figure labels for Fig. 5 and 6 in SD checklist, but the files still have the old labels.

9) As part of the EMBO Publications transparent editorial process initiative (see our Editorial at

<http://embomolmed.embopress.org/content/2/9/329>), EMBO Molecular Medicine will publish online a Review Process File (RPF) to accompany accepted manuscripts. This file will be published in conjunction with your paper and will include the anonymous referee reports, your point-by-point response and all pertinent correspondence relating to the manuscript. Let us know whether you agree with the publication of the RPF and as here, if you want to remove or not any figures from it prior to publication.

10) Please provide a point-by-point letter INCLUDING my comments as well as the reviewer's reports and your detailed responses (as Word file).

I look forward to reading a new revised version of your manuscript as soon as possible.

Yours sincerely,

Poonam Bheda, PhD
Scientific Editor
Molecular Systems Biology

Click on the link below to submit your revised paper.

Reviewer #1:

The authors have addressed all my concerns

Reviewer #2:

The authors have clearly put in a lot of work in response to reviewer comments and addressed all my concerns.

1) In the main manuscript file, please do the following:

- Please include keywords (up to max. 5).

We added the following keywords to the manuscript file:

AlphaFold, protein interaction interface prediction, linear motifs, benchmarking, experimental validation

- Data availability: The Data Availability section needs to be formatted according to the example below:

The datasets and computer code produced in this study are available in the following databases:

- Chip-Seq data: Gene Expression Omnibus GSE46748

(<https://www.ncbi.nlm.nih.gov/geo/query/acc.cgi?acc=GSE46748>)

- Modeling computer scripts: GitHub

(<https://github.com/SysBioChalmers/GECKO/releases/tag/v1.0>)

- [data type]: [full name of the resource] [accession number/identifier] ([doi or URL or identifiers.org/DATABASE:ACCESSION])

Please check "Author Guidelines" for more information.

<https://www.embopress.org/page/journal/17574684/authorguide#availabilityofpublishedmaterial>

We have reformatted the data availability section accordingly.

- Please rename "Conflict of Interest" to "Disclosure and competing interests statement". We updated our journal's competing interests policy in January 2022 and request authors to consider both actual and perceived competing interests. Please review the policy <https://www.embopress.org/competing-interests> and update your competing interests if necessary.

We have revised this section accordingly.

- Author contributions: Please remove it from the manuscript and specify author contributions in our submission system. CRediT has replaced the traditional author contributions section because it offers a systematic machine-readable author contributions format that allows for more effective research assessment. You are encouraged to use the free text boxes beneath each contributing author's name to add specific details on the author's contribution. More information is available in our guide to authors:

<https://www.embopress.org/page/journal/17574684/authorguide#authorshipguidelines>

We have removed the author contributions from the manuscript and ensured that they are correctly entered in the MSB submission system.

- Data not shown: We do not allow statements/conclusions with "data not shown". As per our guidelines, on "Unpublished Data" the journal does not permit citation of "Data not shown". All data referred to in the paper should be displayed in the main or Expanded View figures. Please remove from pages 28 and 35.

The statements have been removed.

2) In the Materials and Methods, please take care of the following:

- Cell lines: Please also be sure to include a sentence in the Materials and Methods as to whether or not the cell lines were recently authenticated.

We have added lines on page 23, lines 33-36 and page 28, lines 22-23 in Materials and Methods to provide information on cell line authentication.

3) Please place individual sections of the manuscript in the following order: Title page - Abstract & Keywords - Introduction - Results - Discussion - Materials & Methods - Data Availability - Acknowledgements - Disclosure and Competing Interests Statement - References - Figure Legends - Expanded View Figure Legends.

We have reordered the sections accordingly.

4) Appendix file and Expanded View content:

- In the Appendix file, please ensure the word "Appendix" is included in all labels for Appendix Figures and Appendix Tables including in the Table of Contents.

We included the word "Appendix" to all labels for Appendix Figures in the Appendix file.

- The Dataset EV1 legend should be removed from Appendix PDF, and included in the zip folder Dataset EV1;

We removed the Dataset EV1 (Now Dataset EV4) legend from the Appendix PDF and included a README file in the zip folder of Dataset EV1 (Now Dataset EV4).

- Tables EV1-EV12 should be renamed to Dataset EV2-EV13 with the corresponding callouts and legends labeled as Dataset EV2-EV13 in each Excel file. These legends should also be removed from the Appendix PDF

We renamed the Tables to Datasets and removed their legends from the Appendix PDF. We have updated all callouts to these Datasets in the manuscript text and figure legends.

5) Data availability: The IM-29904 dataset does not seem to be publicly available. Please be aware that all deposited datasets should be freely accessible prior to publication.

The dataset is now available for download at IntAct under this identifier. However, it is not yet integrated into their current release. You can find the dataset by querying with the above identifier on this website (see also screenshot below):

<https://www.ebi.ac.uk/intact/imex/home.xhtml>

6) Please ensure that all funding sources are entered into the eJP manuscript submission system (i.e. please add the Ministry of Science and Health (MWG),

Rhineland Palatinate (funding ID: TB-Nr.:3658/19); PhD stipend from IMB's collaborative research initiative)

Both missing funding sources have now been added in the submission system.

7) Synopsis: Please check your synopsis text and image before submission with your revised manuscript. Please be aware that in the proof stage minor corrections only are allowed (e.g., typos).

We have checked again the synopsis text and image.

8) Source Data: The panel labels for Source Data files seem to be off. There are updated figure labels for Fig. 5 and 6 in SD checklist, but the files still have the old labels.

We have updated the Source Data file names accordingly.

9) As part of the EMBO Publications transparent editorial process initiative (see our Editorial at <http://embomolmed.embopress.org/content/2/9/329>), EMBO Molecular Medicine will publish online a Review Process File (RPF) to accompany accepted manuscripts. This file will be published in conjunction with your paper and will include the anonymous referee reports, your point-by-point response and all pertinent correspondence relating to the manuscript. Let us know whether you agree with the publication of the RPF and as here, if you want to remove or not any figures from it prior to publication. Please note that the Authors checklist will be published at the end of the RPF.

We agree with the publication of the RPF and do not request removal of any figures.

Reviewer #1:

The authors have addressed all my concerns

Thank you.

Reviewer #2:

The authors have clearly put in a lot of work in response to reviewer comments and addressed all my concerns.

Thank you.

5th Dec 2023

Manuscript number: MSB-2023-11922RR

Title: Systematic discovery of protein interaction interfaces using AlphaFold and experimental validation

Dear Dr. Schueler-Furman,

Thank you again for sending us your revised manuscript. We are now satisfied with the modifications made and I am pleased to inform you that your paper has been accepted for publication.

Yours sincerely,

Poonam Bheda, PhD
Scientific Editor
Molecular Systems Biology
